# Privacy-Preserving Language Model Inference with Instance Obfuscation

## Abstract

Language Models as a Service (LMaaS) offers convenient access for developers and researchers to perform inference using pre-trained language models. Nonetheless, the input data and the inference results containing private information are exposed as plaintext during the service call, leading to privacy issues. Recent studies have started tackling the privacy issue by transforming input data into privacy-preserving representation from the user-end with techniques such as noise addition and content perturbation, while the exploration of inference result protection, namely *decision privacy*, is still a blank page. In order to maintain the black-box manner of LMaaS, conducting data privacy protection, especially for the decision, is a challenging task because the process has to be seamless to the models and accompanied by limited communication and computation overhead. We thus propose Instance-Obfuscated Inference (IoI) method, which focuses on addressing the decision privacy issue of natural language understanding tasks in their complete life-cycle. Besides, we conduct comprehensive experiments to evaluate the performance as well as the privacy-protection strength of the proposed method on various benchmarking tasks.

## 1 Introduction

Language Models as a Service (LMaaS; Yao et al. 2024; Sun et al. 2022; Brown et al. 2020) empowers researchers and developers to access pre-trained language models (PLMs) through cloud services without worrying about the complexities of model training, deployment, and infrastructure management. To interact with LMaaS, users usually send API requests to the designated endpoints designed by the service providers and receive responses generated by the remote language models. Such a setup benefits both parties: on the one hand, users can jump-start on integrating the powerful PLMs into their data processing tasks; on the other hand, the underlying models and the processing pipelines, as the intellectual properties, are hidden from end users so that the service providers can protect them from leakage. However, given the lack of user control over the blackbox cloud service, the data in the requests can be illegally used by the service providers or potential attackers, thus causing privacy issues, including data leakage, unauthorized data access, profiling, and tracking (Sen, 2015; Tang et al., 2016).

Recent literature (Das et al., 2024) has started to address the privacy issues of user inputs in LMaaS, for which solutions are typically based on techniques privatizing the input representation into intermediate ones. Methods of such kind include noise injection (Plant et al., 2021), differential privacy (DP) (Hoory et al., 2021; Yue et al., 2021; Xu et al., 2020), and adversarial training (Li et al., 2018; Coavoux et al., 2018). Moreover, the intermediate representations are further fused or manipulated to prevent reverse engineering, while still remaining sufficient information for effective model inference (Zhou et al., 2022). Unfortunately, to the best of our knowledge, none of the existing methods takes decision privacy into consideration, that is, the inferencing results are not protected which could implicitly or explicitly reveal users' sensitive information based on the specific tasks applied (Shejwalkar et al., 2021; Kahla et al., 2022). For example, as shown in Figure 1, a PLM employed by the online disease diagnosis service can analyze and determine the type of diseases based on the symptom descriptions from the patients. Even though privacy-preserving representations as the input can somehow protect the patients' submitted content, sensitive information such as the distribution of diseases (Mao et al., 2011) from the output aggregation still discloses to the malicious cloud service providers or the

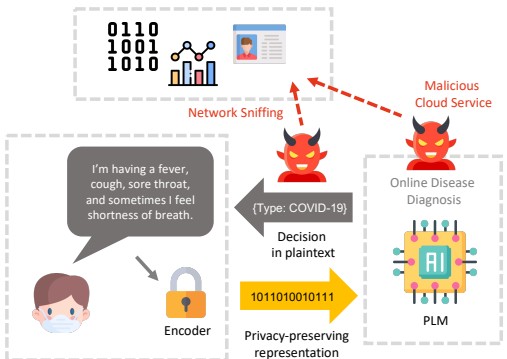

Figure 1: A privacy adversary example with state-of-the-art privacy protection in LMaaS. Despite encoding the end user's input into privacy-preserving representations, the raw output representations or decisions are still in plaintext, making them vulnerable to attacks from both network channels and servers.

hackers via network sniffing.[1] Besides, there are many everyday scenarios where LMaaS is used, including personal assistants, customer support chatbots, language translation, and financial advisory. In almost all these cases, users want to access the service without having their queries and results stored or used for learning.

Considering the significance and necessity of decision privacy protection, we propose to investigate a method that ensures the protection of both raw input content and raw output representation. However, protection in the decision phase could be more challenging than its counterpart in the input phase due to several reasons. First, unlike user inputs, since the final decision is made by the PLM on the cloud, users will have no direct means to intervene in it. Second, due to the required anonymity, incurred communication costs inevitably increase. Third, from the perspective of intellectual property protection, it is not practical for LMaaS providers to disclose parameters and architectures of the models, including the last few layers that are close to the decisions, to users. These challenges call upon a solution that effectively protects the models' decisions before they come out, while does not violate the black-box nature of the LMaaS.

In this paper, we propose IoI (Instance-obfuscated Inference), which aims to protect the privacy of PLM decisions without losing the compatibility of utilizing the state-of-the-art input privacy protection approaches at inference phase. During inference, IoI intentionally obfuscates the instance, hiding the raw decision distribution from revealing any sensitive information. However, the user who applies the obfuscation retains the ability to recover the true decision distribution. Note as a pilot study, IoI focuses on text classification tasks.

Despite distinctiveness, to avoid the ambiguity of understanding different privacy techniques, in Figure 2, we summarize them according to the application scenarios. Specifically, SOTA methods utilize DP for training time data privacy. Other aforementioned noise addition or perturbation methods safeguard the raw input from being reverse-engineered. On the contrary, IoI ensures the confidentiality of decisions from the model inference.

The contributions of this work are three-folds. First, we explore the feasibility of protecting PLM *decisions* in *a black-box manner* (operating solely during inference without requiring any training) for text classification tasks. Second, we define *decision privacy*, and comprehensively study the instance obfuscation strategies and privacy-preserving decision resolution in the context of it. Third, we define evaluation metrics for decision privacy, and empirically verify the performance and privacy strength of the proposed method.

## 2 Privacy-Preserving Inference

For a text classification task $M : \mathcal{X} \to \mathcal{Y}$, where $\mathcal{X}$ is the input text and $\mathcal{Y}$ is the label set. The *privacy-preserving inference* takes a step further avoiding the exposure of any private information about the inputs

---

[1]The security of the network channel is not the scope of this paper. You can assume it is already end-to-end encrypted.

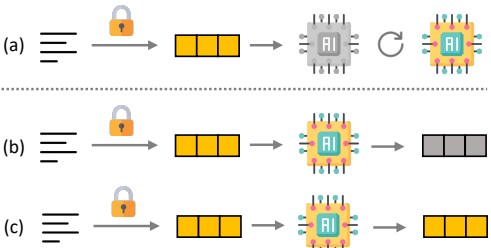

Figure 2: Privacy-preserving scenario comparison. **(a) Training Privacy** aims to protect the private training data. A typical privacy tool for this scenario is differential privacy. **Inference Privacy** includes **(b) Input Privacy** that prevents the raw input data from being revealed; and **(c) Decision Privacy** that protects the inference results. The vectors in orange are privacy-preserving, while the ones in gray are not.

and model decisions to the service provider. While the encoding methods[2] for protecting the privacy of $\mathcal{X}$ start emerging, the counterpart for $\mathcal{Y}$, which we call decision privacy, remains uncharted.

The intuition for achieving decision privacy is to make the model's raw decision as random as possible to all parties except the input instance owner, and the raw decision can only be recovered via a certain resolution method by the input instance owner. In the rest of this section, we formally define decision privacy in the context of text classification, as well as privacy-preserving inference.

## 2.1 Decision Privacy

For text classification, suppose $(x, y)$ is an instance of $(\mathcal{X}, \mathcal{Y})$, and a finite label set $C = \{c_i | 1 \leq i \leq n, n \geq 2\}$ is the range of $\mathcal{Y}$. We say $M$'s output has *perfect privacy* if

$$Pr[M(x) = c_i] \approx \frac{1}{n}, \tag{1}$$

that is, the probability of an adversary acquiring the predicted label $c_i$ from $M$ for the given input $x$ is almost *no better than a random guess*.

However, directly adhering to Equation (1) leads to compromised functionality of $M$, since $M$ is essentially a random choice function and useless in practice. Instead, a certain encoding function $E(\cdot)$ can be performed on the input $x$, so that the decision privacy of an arbitrary model $M$ is ensured by $E(\cdot)$:

$$|Pr[M(E(x)) = c_i] - \frac{1}{n}| \leq \epsilon, \tag{2}$$

where $\epsilon \in [0, 1)$ is seen as a privacy budget. Adjusting $\epsilon$ balances the utility and privacy: the smaller the $\epsilon$ is, the better the decision privacy.

## 2.2 Problem Definition

The privacy-preserving inference is defined as:

$$M(E(x)) \rightarrow y', \tag{3}$$

where the encoding function $E(\cdot)$ has **two** functionalities: (1) It encodes the raw $x$ into some privacy-preserving representation remaining interpretable by $M$, which is already studied by previous work (Qu et al., 2021; Yue et al., 2021; Zhou et al., 2022) and is **not** the focus of this paper. (2) The inference result

---

[2]The encoding mentioned in this paper is not by the PLM's encoder but as "encryption".

transitions from the actual prediction $y$ to the privacy-preserving $y'$, whose distribution satisfies decision privacy defined by Equation (2):

$$|Pr[y' = c_i] - \frac{1}{n}| \leq \epsilon. \tag{4}$$

The privacy property is ensured by $E(\cdot)$. It is mathematically hard or impossible to find its inverse function $E^{-1}(\cdot)$, so that the adversary can not either recover *the raw input $x$* from the privacy-preserving representation $E(x)$, or *the actual prediction $y$* from the privacy-preserving prediction $y'$. A decoding function $D(\cdot)$ is available to decode true $y$ from $y'$ with the knowledge of the raw input and encoding settings,

$$y \leftarrow D(y', E, x). \tag{5}$$

Without loss of generality, rather than determining the true prediction $y$ from a single $y'$, Equation (5) can be extended to cases where the solution for $y$ depends on multiple $y'$ values, that is,

$$y \leftarrow D(y'_0, \cdots, y'_g, E, x), \tag{6}$$

where $y'_0 \cdots y'_g$ are all necessary $y$'s to decode $y$. Note that $E$ and $x$ serve to identify these $y'$ values as a group and can be omitted if the user maintains the reference between $y'$s and $y$.

Therefore, privacy-preserving inference allows the user to query LMaaS without exposing sensitive information to the service provider or the adversary, by sending $E(x)$ to the server and decoding $y$ from $y'$ with $D(\cdot)$ locally. Throughout the process, *the adversary learns nothing from the encoded input $E(x)$ or encoded decision $y'$*.

**Distinctions to DP or input privacy.** In general, DP adds proper noise to the given input instances so that the individual information of input instances will not be leaked, but the overall statistical features of them remain. Similarly, most input privacy methods perturb the raw input to prevent input reverse-engineering while keeping the necessary information for inference. Hence, an ideal DP or input privacy method should satisfy $M(E(x)) \approx M(x)$, where $E$ is the corresponding DP or input privacy method, while protecting the privacy of $x$. On the contrary, decision privacy tends to make $M(E(x))$ as random as possible whereas $D(M(E(x))) \approx M(x)$.

## 3 Method

This section begins with an overview of our privacy-preserving inference framework for text classification. It follows by detailing the core component $E(\cdot)$ for encoding in Section 3.1, and $D(\cdot)$ for decoding in Section 3.2.

The intuition behind IoI is to obfuscate the raw instance with obfuscators so that the PLM's inference distribution is intentionally steered. Thus, the adversaries cannot deduce the true decision unless they possess knowledge of the corresponding resolution method and parameters. The general workflow of IoI is shown in Figure 3 (we also provided an end-to-end pseudo-code in Appendix A.1), consisting of instance obfuscation as $E(\cdot)$ and decision resolution as $D(\cdot)$. Instead of sending the raw instance $x$ to the PLM and acquiring the decision, IoI conceals $x$ by concatenating it with an obfuscator $b$, which is also a text sequence (Section 3.1). The concatenated text $[b; x]$ along with the obfuscator $b$ are sent to the privacy-preserving representation generation (PPRG) module, respectively, where the input is encoded by a compatible SOTA input privacy method. PPRG produces privacy-preserving representations, which are irreversible and remain distinct even for identical inputs, and are treated as inputs for the PLM. After PLM's inference on PPRG-encode $b$ and $[b; x]$, the raw decision distribution of $[b; x]$ does not reflect the inference of $x$ since it is steered by the elaborated obfuscator $b$. But as the data owner, the actual decision $y$ can be resolved via decision resolution (Section 3.2) by utilizing the decision distributions of $[b; x]$ and corresponding $b$. We further show that the true $y$ is hard to be recovered from $y'$s in Section 4.

### 3.1 Instance Obfuscation

Sending the input instance $x$ in plaintext to the PLM reveals the input completely. Hence, some previous studies (Zhou et al., 2022; Plant et al., 2021) employ a *privacy-preserving text representation* that transforms

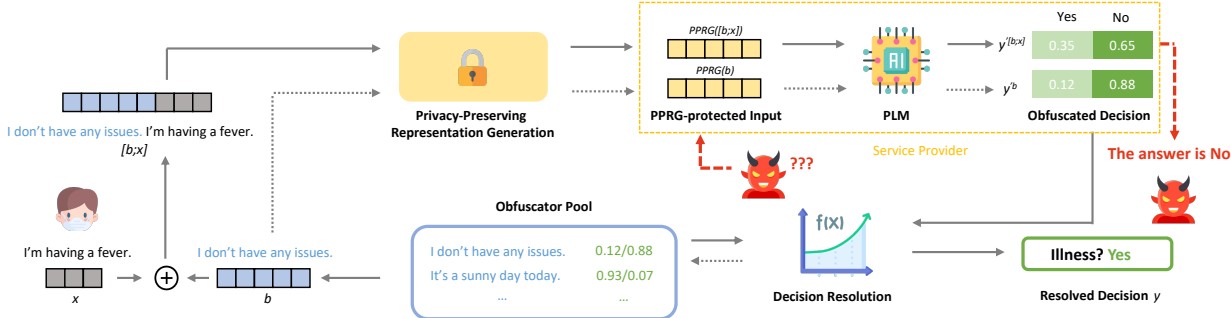

Figure 3: The demonstration of IoI workflow for decision privacy protection. If a user (bottom left) makes illness inquiries via a PLM-driven online diagnosis system, normally, the inference result will be returned in plain text. As a most basic example, in IoI, the raw text is concatenated with an obfuscator, which is also a text. Subsequently, the concatenated text and the obfuscator are encoded respectively by the privacy-preserving representation generation module, which ensures the produced embedding representation is privacy-preserving (irreversible and unique). Consequently, instead of receiving one "plaintext", the PLM receives two independent "ciphertext" and makes inferences on them without knowing their correlation, raw text, and true decision. However, only the user is able to recover the true decision by leveraging the distribution of these two inferences. In practice, each input text is obfuscated by a group of obfuscators, and the requests from multiple inputs are sent to the PLM in arbitrary order.

the input into "ciphertext" form by perturbing representations. In this way, although the raw input content is not exposed, the output of the PLM still carries meaningful information and may be exploited by the adversary. To tackle the flaw of limited protection of the decision by previous methods, IoI uses instance obfuscation, acting as $E(\cdot)$ in Section 2. It not only protects the input privacy by reusing the existing SOTA privacy-preserving text representation methods as PPRG but also "fools" the adversary with baffled output for decision privacy.

The instance obfuscation is motivated by mixup (Zhang et al., 2018), originally proposed for data augmentation. Zhang et al. (2018) shows that mixup can produce virtual feature-target pairs sampled from the same distribution as the original data. Specifically, it shows that, through a mapping (i.e., the LM in this context), the mixup of two raw inputs can be mapped to the mixup of their corresponding labels. Based on that, if $E(\cdot)$ conceals the real instance $x$ by mixing it up with dummy instances, the PLM only makes an inference on the mixup instance without seeing $x$, and the proportion of dummy instances participated in the mixup steers the final decision. We call these dummy instances obfuscators $b$; thus, $E(\cdot)$ obscures the true instance with selected obfuscators and let the PLM make decisions based on the elaborated input. However, our initial experiments indicate that constructing $E(\cdot)$ by directly mixing embedding vectors of $x$ and $b$, following the original approach of Zhang et al. (2018), results in unstable performance for $D(\cdot)$. This instability arises because a vanilla PLM without mixup fine-tuning may lack the ability to accurately infer the mixed label solely from the mixed input. Therefore, we replace the proportional mixup of $b$ and $x$ with concatenation. Namely, each $x$ is represented by obfuscators $b$s and obfuscated instances $[b;x]$s

$$x \equiv \{[b_i;x]\}_{i=1}^k \cup \{b_i\}_{i=1}^k, \tag{7}$$

where ; denotes the text concatenation, $\equiv$ means "represented by", and $k$ is the number of obfuscators.

The obfuscation process is the key for concealing information in a black-box LMaaS setting. This leads to the question of what is considered to be a high-quality obfuscator and how to obfuscate $x$ with $b$ properly, hence maximizing the performance as well as privacy protection.

**Obfuscator Selection.** Obfuscators are simply normal (unlabeled) sentences that could be with or without any relation to the real instances to be protected. To be used as an obfuscator, an instance requires to have

a corresponding predicted label from PLM. Note that the predicted label does not need to be correct so there is no need for a gold label. Thus, an obfuscator $b$ could be a sentence from any arbitrary corpus.

To steer the PLM's decision towards being affected by $b$ instead of $x$, we prioritize $b$ instances with higher confidence regarding the PLM decision. For example, in a binary classification task, if an instance $x_1$ scored 0.9 for label 1 and $x_2$ scored 0.7, then $x_1$ is picked over $x_2$ for $x_1$ is more deterministic to get label 1. Since the selected obfuscators can be paired with any real instances, an optimized way to re-use them is to have them pre-computed in an *obfuscator pool*.

**Obfuscator Balancing.** Based on the observation, a single $b$ instance for obfuscating $x$ results in the un-stableness in decision resolution (Section 3.2) due to the uneven distribution of the PLM's decision. For example, in a 3-class classification, assume $b_1$ has label $c_1$, $b_2$ has label $c_2$, and $b_3$ has label $c_3$. After a single obfuscation with $b_1$, the label of $[b_1; x]$ predicted by $M$ could remain $c_1$, or change to $c_2$ or $c_3$. Thus, the steering of decision distribution using a single obfuscator is not steady. Balancing, as a solution, is employed to mitigate this issue. Specifically, each real instance $x$ is paired with *at least* one *unit group* of obfuscators. A unit group of obfuscators is defined as a set containing obfuscators with uniformly distributed labels from the label set $C$, that is,

$$g = \{b_j \in B \mid M(b_j) = c_i, \forall c_i \in C\} \text{ with } |g| = |C|, \tag{8}$$

where $B$ is the obfuscator pool. This means a $|C|$-size $g$ contains *one qualified combination* of $b_j$s. Moreover, to enhance the balancing effect, a group can consist of more than one unit group. Formally, a group contains $n$ unit groups is defined as

$$G_n = g_1 \cup g_2 \cup \cdots \cup g_n. \tag{9}$$

Therefore, the obfuscated instances of $x$ are noted as $[b_i; x]$ where $b_i \in G_n$. Using balancing in the previous example, assume that only a unit group of obfuscators is used ($n = 1$) for obfuscation; then $x$ should concatenate with all three obfuscators and result in three obfuscated instances $[b_1; x]$, $[b_2; x]$ and $[b_3; x]$.

**Privacy-Preserving Representation Generation.** Even though the raw instance $x$ is replaced with $[b; x]$ and $x$, the content remains in plaintext. To protect their privacy, IoI uses PPRG, which can be *any compatible SOTA input privacy methods* (Zhou et al., 2022; Plant et al., 2021), for transforming $[b; x]$ and $b$ from text sequences to privacy-preserving representations. A qualified SOTA input privacy method has two requirements regarding privacy. First, the produced representation is not invertible so that the adversary can not reverse it back to plaintext. Second, the input privacy method is equipped with randomness so that the produced representation is distinct even for identical inputs. As long as the methods satisfy these two requirements and convert input text to embedding vectors, they are considered compatible. After applying PPRG, the order of multiple $[b; x]$s and $b$s should be *uniformly shuffled* and sent to PLM. This prevents the adversary from pairing up the encoded $[b; x]$ and $b$. A more detailed discussion regarding privacy is in Section 4.

In conclusion, the process of encoding $E(\cdot)$ on one instance $x$ can be formally denoted as

$$E(x) \equiv \{\mathsf{PPRG}([b_i; x])\}_{i=1}^{|G_n|} \cup \{\mathsf{PPRG}(b_i)\}_{i=1}^{|G_n|}. \tag{10}$$

For $k$ input instances $\{x_i\}_{i=1}^{k}$, after encoding, the representation is

$$\{E(x_i)\}_{i=1}^{k} \equiv \sigma(\{\mathsf{PPRG}([b_i; x])\}_{i=1}^{k \times |G_n|} \cup \{\mathsf{PPRG}(b_i)\}_{i=1}^{k \times |G_n|}), \tag{11}$$

where $\sigma$ denotes the uniform shuffle applied to a set.

### 3.2 Privacy-Preserving Decision Resolution

While the obfuscated instance ceases the raw instance $x$ from being accessible by the PLM, the true decision of $x$ is concealed in the concatenated result $y'$s as well. We outline a decision resolution method, as $D(\cdot)$ in Section 2 (Equation (6)), to resolve true $y$ from multiple associated $y'$s.

As the balancing described in Section 3.1, successfully executing $D(\cdot)$ to get the decision of $x$ requires all the associated $[b; x]$ and $b$ pairs. As to adversaries, correctly locating all associated instances from a tremendous amount of mixed instances, which are sufficiently obfuscated and randomized, is equivalent to finding a needle in a haystack. We have detailed analysis in Section 4.

Oppositely, as the data owner, running $D(\cdot)$ is as easy as pie. The strategy to separate $x$'s result $y$ from $y'$s is based on the divergence between the decision distribution of $[b; x]$ and $b$. Specifically, if $x$'s label is $c_k$, blending it with $b$ shifts the confidence of $[b; x]$'s decision distribution towards $c_k$ regardless of the $b$'s label. Taking our example in Figure 3, $[b; x]$ has 0.35 for "yes" and 0.65 for "no," while $b$ has 0.12 for "yes" and 0.88 for "no". The confidence of "yes" for $[b; x]$ increases because of the involvement $x$, thus $x$ is highly like to be "yes."

Without loss of generality, we inductively evaluate this divergence over a group $G_n$. The true label of $x$ is determined as the one for which the average confidence difference between the decision of the obfuscated instance and the obfuscator exhibits the greatest variation. Formally, the true label is determined as

$$\arg\max \frac{1}{|G_n|} \sum_{j=1}^{|G_n|} (\boldsymbol{z}^{[b_j;x]} - \boldsymbol{z}^{b_j}), \tag{12}$$

where $b_j \in G_n$. The logits of the model decision are denoted by $\boldsymbol{z}$, represented as a $|C|$-dimensional vector, i.e., $\boldsymbol{z} \in \mathbb{R}^{|C|}$. The superscripts in $\boldsymbol{z}^{[b_j;x]}$ and $\boldsymbol{z}^{b_j}$ indicate that $\boldsymbol{z}$ is inferred from the corresponding input.

In conclusion, the process of decoding $D(\cdot)$ on $k$ instance $x$s can be formally denoted as

$$\{D(E(x_i))\}_{i=1}^k = \{\arg\max \frac{1}{|G_n|} \sum_{j=1}^{|G_n|} (M(\mathsf{PPRG}([b_j; x_i]) - M(\mathsf{PPRG}(b_j))\}_{i=1}^k. \tag{13}$$

Note that, except for the data owner, no one can distinguish whether an input to $M$ originates from $[b; x]$ or $b$, as they are PPRG-encoded. Additionally, the association between $\boldsymbol{z}$s and a specific $x$ remains unknown, since the inputs to $M$ are uniformly shuffled. A more detailed discussion on privacy is provided in Section 4.

## 4 Privacy Discussion

In this section, we formally define the threat model in Section 4.1, and rigorously prove IoI's privacy in Section 4.2. Moreover, we discuss the incurred cost of privacy in Section 4.3.

### 4.1 Threat Model and Privacy Definitions

**Definition 1** *(Honest-but-curious adversary) An honest-but-curious adversary is a proper, but passive participant of a communication protocol who does not deviate from it, but attempts to learn as much information as possible from all legitimate communication (Paverd et al., 2014).*

**Definition 2** *(Irreversibility) A function $E$ is irreversible if no probabilistic polynomial-time adversary $\mathcal{A}$ can efficiently compute the input $x$ given only $E(x)$. Formally, for any such $\mathcal{A}$, the advantage*

$$\Pr[\mathcal{A}(E(x)) = x] \leq \epsilon(\theta)$$

*for a negligible function $\epsilon$ in the security parameter $\theta$.*

**Definition 3** *(Distinctiveness) A function $E$ is said to be* distinctive *if, for any input $x$, the outputs $E(x)$ are distinct even when the same input is provided multiple times. Formally,*

$$\Pr[E(x_1) = E(x_2)] \leq \epsilon(\theta),$$

*where $x_1 = x_2$ and $\epsilon(\theta)$ is a negligible function of the security parameter $\theta$.*

**Definition 4** *(Security) A protocol $\Pi$ is said to be* secure *if it satisfies the following condition: there exists a simulator $\mathcal{S}$ such that no environment $Z$ can distinguish between an execution of the protocol $\Pi$ and an ideal execution with the ideal functionality $F$. Formally, for any adversary $\mathcal{A}$ and any environment $Z$, the following condition must hold:*

$$|\mathsf{Real}_{\Pi,\mathcal{A},Z} - \mathsf{Ideal}_{F,\mathcal{S},Z}| \leq \epsilon(\theta),$$

*where $\mathsf{Real}_{\Pi,\mathcal{A},Z}$ denotes the view of $Z$ in a real execution of the protocol $\Pi$, involving the adversary $\mathcal{A}$; $\mathsf{Ideal}_{F,\mathcal{S},Z}$ denotes the view of $Z$ in an ideal execution with the ideal functionality $F$ and the simulator $\mathcal{S}$, who acts on behalf of the adversary; $\epsilon$ is a negligible function in the security parameter $\theta$ (Canetti, 2001).*

We assume the presence of an honest-but-curious (semi-honest) adversary (Definition 1) $\mathcal{A}$ who can be the service provider $P_2$ (having white-box access to the PLM $M$) or any other entity that eavesdrops on the inputs and outputs of $P_2$.

Given the inherent uncertainty in language models (arising from their probabilistic nature, data ambiguity, overparameterization, prompt sensitivity, decoding strategies, etc), we introduce an additional parameterization to quantify the security as follows:

**Definition 5** *($(\epsilon,\delta)$-security) A protocol is said to be $(\epsilon,\delta)$-secure if it satisfies Definition 4 under the $(\epsilon,\delta)$ condition. The parameter $\epsilon$ quantifies the tolerance for output randomness (Equation (2)). The parameter $\delta$ bounds the difference in the inference logit distribution between $[b;x]$ and $b$, formally defined as $D_{\mathsf{KL}}(P(\boldsymbol{z}^{[b;x]}) \parallel P(\boldsymbol{z}^b)) + D_{\mathsf{KL}}(P(\boldsymbol{z}^b) \parallel P(\boldsymbol{z}^{[b;x]})) \leq \delta$, where $D_{\mathsf{KL}}$ denotes the KL divergence (Kullback and Leibler, 1951).*

## 4.2 Simulation-based Proof

The IoI protocol involves two parties, a client $P_1$ and a service provider $P_2$. The client $P_1$ aims to query the remote PLM $M$ (in $P_2$) with $k$ text instances $x$s while ensuring that the original inputs $x$s and the corresponding decisions $y = M(x)$s remain private from any other parties.

We use a simulation-based proof (Lindell, 2017) to show that the real-world execution is indistinguishable from an ideal-world functionality. The **ideal-world trusted functionality** $\mathcal{F}$ is specified in Figure 4. The detailed IoI's protocol is specified in Algorithm 2, and can be simplified to the **real-world protocol** $\pi$ in Figure 5 for privacy analysis. We describe a **simulator $\mathcal{S}$** that simulates the view of the $\mathcal{A}$ in the real-world execution of $\pi$.

1. $P_1$ sends $k$ text instances $x$ to $\mathcal{F}$.
2. $P_2$ sends the PLM $M$ to $\mathcal{F}$.
3. $\mathcal{F}$ computes $y = M(x)$ and sends $y$ to $P_1$.
4. $P_2$ receives nothing.

Figure 4: IoI's ideal-world functionality $\mathcal{F}$

Mathmatically, the view in the ideal-world is $\mathsf{View}^{\mathcal{F}} = \emptyset$, and the view in the real-world is $\mathsf{View}^{\pi} = \{\mathsf{PPRG}([b_i;x])\}_{i=1}^{k \times |G_n|} \cup \{\mathsf{PPRG}(b_i)\}_{i=1}^{k \times |G_n|} \cup \{y_i'\}_{i=1}^{2 \times k \times |G_n|}$.

1. **Local input encoding:** Each original instance $x$ is encoded to a privacy-preserving representation $E(x)$ by $P_1$. Specifically, $E(\cdot)$ contains two steps:

    i. Each $x$ is transformed into two sets of instances:

$$\{[b_1; x], \cdots, [b_{|G_n|}; x]\} \quad \text{and} \quad \{b_1, \cdots, b_{|G_n|}\}.$$

    ii. Both sets are then processed using a compatible PPRG method:

$$\{\mathsf{PPRG}([b_1; x]), \cdots, \mathsf{PPRG}([b_{|G_n|}; x])\} \quad \text{and} \quad \{\mathsf{PPRG}(b_1), \cdots, \mathsf{PPRG}(b_{|G_n|})\}.$$

Additionally, $P_1$ uniformly shuffles the sending order of all $k \times 2 \times |G_n|$ encoded instances and maintains a list $O$ which records each $x$ along with its corresponding $[b; x]$s and $b$s.

2. **Remote PLM inference:** The encoded and shuffled instances $E(x)$s are transmitted to the service provider $P_2$ for inference, where each encoded input is processed as $y' = M(E(x))$.

3. **Local decision resolution:** Upon receiving $k \times 2 \times |G_n|$ outputs $y'$s, the client groups them into $k$ sets according to $O$. For each group of $y'$s, client executes decision resolution to acquire the true decision

$$y = D(y's) = D(y'^{[b_1; x]}, \cdots, y'^{[b_{|G_n|}; x]}; y'^{b_1}, \cdots, y'^{b_{|G_n|}}).$$

Figure 5: IoI's real-world protocol $\pi$

$\mathcal{A}$ can challenge Definition 4 in two aspects: (1) reverse the original input $x$ from $E(x)$; (2) recover the true output $y$ from all available obfuscated decisions $y'$s.

**Reversing original input.** To reverse $E(x)$ to $x$, $\mathcal{A}$ needs to reverse the representation produced by PPRG. Then $\mathcal{A}$ identifies $b$s and $[b; x]$s, and extracts $x$ from one of the $[b; x]$s.

As mentioned in Section 3.1, a qualified PPRG ensures the *irreversibility* (Definition 2) of the input text sequence, meanwhile generating *distinct* (Definition 3) representations even for identical input. Although $\mathcal{A}$ may have white-box access to $M$, these do not help $\mathcal{A}$ reverse PPRG encoded instances as long as PPRG has sufficient privacy strength to resist known attacks (Song and Raghunathan, 2020). Note that, from the perspective of $\mathcal{A}$, it cannot determine which is $\mathsf{PPRG}([b; x])$ and which is $\mathsf{PPRG}(b)$, i.e., $\mathsf{View}^\pi = \{\mathsf{PPRG}(x'_i)\}_{i=1}^{2 \times k \times |G_n|} \cup \{y'_i\}_{i=1}^{2 \times k \times |G_n|}$, where $x'$ can either be $[b; x]$ or $b$. Therefore, a $E(x)$ is computationally indistinguishable from a random string.

In the simulation, $\mathcal{S}$ generates a random string $r$, which has the same length as $E(x)$. Thus, we get $\mathsf{View}^{\mathcal{F}} = \{r_i\}_{i=1}^{2 \times k \times |G_n|}$.

**Recover true decision.** When resolving $y$ from $y'$s without knowing $O$, $\mathcal{A}$ must determine three aspects: (1) identify all $|G_n|$ associated $y$ values from the $k \times 2 \times |G_n|$ available $y'$s, (2) distinguish which $y'$ originates from $[b; x]$ and which from $b$, (3) verify whether the inputs of two $y'$s form a pair, meaning $[b; x]$ and $b$ share the same $b$. However, it is no better than exhausting all the possible combinations because of the hardness of reversing PPRG-encoded instances and the uniformly shuffled order of unique representations. Therefore, all the $y'$ is also computationally indistinguishable from a random string.

In the simulation, $\mathcal{S}$ generates a random string $s$, which has the same length as a $y'$. Thus, we get $\mathsf{View}^{\mathcal{F}} = \{r_i\}_{i=1}^{2 \times k \times |G_n|} \cup \{s_i\}_{i=1}^{2 \times k \times |G_n|}$.

The execution of $\mathcal{S}$ implies $\mathsf{View}^{\mathcal{F}} \equiv \mathsf{View}^{\pi}$, which satisfies Definition 5. Since no computationally bounded $\mathcal{A}$ can distinguish the real-world protocol from the ideal-world functionality, we prove that the protocol is secure under simulation-based security.

### 4.3 Privacy Cost

Privacy comes with a cost. Here, we elaborate it from two aspects: communication and computation.

**Communication cost.** As a baseline, each instance $x$ sends one request to the PLM. In IoI, as Equation (9), each $x$ is concatenated with $|G_n| = n \times |C|$ instances. All these obfuscated instances, along with the same amount of obfuscators, form the total requests to the PLM, namely, $2 \times n \times |C|$. In practice, when there are multiple $x$s, the obfuscators are pre-computed and reused from the obfuscator pool. Hence, for $k$ $x$ instances, the total number of requests ranges in $[(1 + k \times n) \times |C|, 2 \times k \times n \times |C|]$.

**Computation cost.** On PLM's side, the number of requests to inference is indicated in the communication cost. On the data owner's side, resolving a $y$ requires the execution of Equation (12), which only involves trivial matrix operations.

## 5 Experiments

We first introduce the datasets, baselines, and evaluation metrics in Section 5.1. The main results and multilingual results regarding task performance and decision privacy are illustrated in Section 5.2. We further study the functionalities of technical components in Section 5.3.

### 5.1 Experimental Setup

**Datasets.** Our experiments are conducted on four benchmark datasets that span across various text classification tasks. **SST-2** (Socher et al., 2013) requires to classify the sentiment of the given text into either positive or negative class. **SST-5**, as an extension of the SST-2, granularizes the binary label into five categories: very negative, negative, neutral, positive, and very positive. **MRPC** (Dolan and Brockett, 2005) is a paraphrase identification task to determine whether two sentences are paraphrases. **QNLI** (Wang et al., 2018), derived from SQuAD (Rajpurkar et al., 2016), is a natural language inference task seeking to identify if the context sentence contains the answer to the question. **Financial Phrasebank (Fin)** (Malo et al., 2014) is for English language financial news sentiment classification. **Tweet Sentiment Multilingual (TSM)** (Barbieri et al., 2022) consists of sentiment analysis dataset on Twitter in 8 different languages.

**Baselines.** Although no direct comparable methods regarding decision privacy are available, we select four reasonable and related baselines. **Fine-tuned** is task fine-tuned model without privacy protection. **Random Guess** denotes the random guess results. **PP-BERT** (Qu et al., 2021) is a privacy-preserving encoder that perturbs the token embeddings by adding random noise $N = rp$ where $r$ is the distance from the origin and $p$ is a unit hypersphere. $r$ is sampled from the Gamma distribution $\Gamma(n, \frac{1}{\eta})$.[3] **SanText+** (Yue et al., 2021) replaces sensitive words with GloVe (Pennington et al., 2014) and utilizes differential privacy to ensure the privacy of the sanitized words.[4] **TextFusion** (Zhou et al., 2022) alters the input text sequence or intermediate representations by eliminating redundant or sensitive information. Note PP-BERT, SanText+, and TextFusion were all designed for input privacy.

**Metrics.** Generally, the performance is evaluated by task-specific metrics (Accuracy/F1) denoted as $T$. To measure the raw performance, we use the obfuscated ($T_o$) for the obfuscated version of the decision, and the resolved ($T_r$) for the true (original) performance from the decision resolution.

Besides, to quantify the effectiveness of the decision privacy protection and decision resolution, we additionally define $\Phi_r = \frac{T_{\text{baseline}} - T_r}{T_{\text{baseline}}}$ and $\Phi_o = \frac{|T_o - T_{\text{random}}|}{1 - T_{\text{random}}}$. They measure the relative performance difference

---

[3] We set $\eta = 100$, a moderate value in the original paper, for balancing the noise strength and accuracy.

[4] Considering fairness, we set $\epsilon = 3$ in SanText+ and use sanitized text directly in inference.

| Dataset | Method | $T_r$ | $T_o$ | $\Phi_r \downarrow$ | $\Phi_o \downarrow$ | $\Phi \downarrow$ |
|---|---|---|---|---|---|---|
| SST-2 (Acc.) | Fine-tuned | .924 | .924 | − | − | − |
| | Random Guess | .500 | .500 | − | − | − |
| | PP-BERT | .909 | .909 | .016 | .818 | .417 |
| | SanText+ | .830 | .830 | .102 | .660 | .381 |
| | TextFusion | .904 | .904 | .022 | .808 | .415 |
| | IoI | .913 | .770 | **.012** | **.540** | **.276** |
| MRPC (Acc./F1) | Fine-tuned | .860/.904 | .860/.904 | − | − | − |
| | Random Guess | .500/.500 | .500/.500 | − | − | − |
| | PP-BERT | .434/.294 | .434/.294 | .489/.675 | **.132**/.412 | .310/.543 |
| | SanText+ | .711/.750 | .711/.750 | .164/.170 | .422/.500 | .293/.335 |
| | TextFusion | −/.882 | −/.882 | −/**.024** | −/.764 | −/.394 |
| | IoI | .745/.794 | .570/.628 | **.124**/.122 | .166/**.256** | **.132**/**.189** |
| SST-5 (Acc.) | Fine-tuned | .500 | .500 | − | − | − |
| | Random Guess | .200 | .200 | − | − | − |
| | PP-BERT | .490 | .490 | **.020** | .362 | .191 |
| | SanText+ | .426 | .426 | .148 | .282 | .215 |
| | IoI | .467 | .339 | .066 | **.174** | **.120** |
| QNLI (Acc.) | Fine-tuned | .915 | .915 | − | − | − |
| | Random Guess | .500 | .500 | − | − | − |
| | PP-BERT | .658 | .658 | .281 | .316 | .298 |
| | SanText+ | .725 | .725 | .208 | .450 | .329 |
| | IoI | .849 | .648 | **.072** | **.296** | **.184** |
| Fin (Acc.) | Fine-tuned | .907 | .907 | − | − | − |
| | Random Guess | .333 | .333 | − | − | − |
| | PP-BERT | .254 | .254 | .720 | **.118** | .419 |
| | SanText+ | .853 | .853 | .060 | .780 | .420 |
| | IoI | .855 | .683 | **.057** | .525 | **.291** |

Table 1: Performance of resolved and obfuscated decisions by IoI and baselines. $T_r$ indicates the task raw performance after decision resolution by the data owner, while $T_o$ indicates the performance that model owner or attacker retrieves. $\Phi_r$ and $\Phi_o$ measure how close $T_r$ and $T_o$ to the baseline and the random guess, respectively, and $\Phi$ is a balance between $\Phi_r$ and $\Phi_o$. The smaller the three $\Phi$s, the better the task performance and decision privacy protection. The best result in each task is highlighted in **bold**. Only IoI has effects on decision privacy, while the other baselines are either non-privacy-preserving or only protect input privacy.

from resolved $T_r$ to non-privacy-preserving baseline, and from obfuscated $T_o$ to random guess, respectively. Finally, a unified metric $\Phi = \frac{\Phi_o + \Phi_r}{2}$ measures the balance between the obfuscation strength and the task performance.[5]

## 5.2 Main Results

The backbone PLM used in each task is fine-tuned and consistent with all baselines because PP-BERT, SanText+, TextFusion, and IoI are applied in the inference phase. The parameter settings of IoI are in Table 2. PPRG is not enabled in this experiment and the effect of it is studied in Section 5.3. As the main results presented in Table 1, IoI performs almost the best among all tasks regarding the resolved ($\Phi_r$) and obfuscated ($\Phi_o$) results, and the balance between them ($\Phi$), except few are lower but close to the best baselines. Note that only IoI can protect decision privacy, thus, its $\Phi_o$ is the best as compared to all others.

Specifically, on SST-2 and QNLI, the resolved results $T_r$ by IoI have similar accuracy as the non-privacy fine-tuned baselines indicated by the smaller $\Phi_r$ while still deviating obfuscated result $T_o$ to be as close to random as possible showing as the smaller $\Phi_o$. For SST-5, as a harder version of SST-2, albeit it is not the best regarding task performance, IoI balances the trade-off, indicated by $\Phi$, to still archive relatively better decision privacy. For MRPC and Fin, the evaluation metrics capture the abnormal performance of

---

[5]All three metrics are scaled to be in the range $[0, 1]$, and the smaller value indicates better performance.

| | SST-2 | MRPC | SST-5 | QNLI | Fin | TSM (ar) | TSM (es) | TSM (fr) | TSM (it) |
|---|---|---|---|---|---|---|---|---|---|
| Max sequence len | 128 | 512 | 128 | 256 | 128 | 128 | 128 | 128 | 128 |
| Min confidence of $b$ | $> 0.99$ | $> 0.90$ | $> 0.90$ | $> 0.99$ | $> 0.90$ | $> 0.95$ | $> 0.92$ | $> 0.90$ | $> 0.95$ |
| Group size $n$ | 1 | 1 | 1 | 1 | 1 | 1 | 1 | 1 | 1 |

Table 2: IoI settings for main and multi-lingual results

| Language | Fine-tuned | Random | $T_r$ | $T_o$ | $\Phi_r \downarrow$ | $\Phi_o \downarrow$ | $\Phi \downarrow$ |
|---|---|---|---|---|---|---|---|
| Arabic (ar) | 0.647 | 0.333 | 0.615 | 0.469 | 0.049 | 0.204 | 0.127 |
| Spanish (es) | 0.706 | 0.333 | 0.541 | 0.417 | 0.234 | 0.126 | 0.180 |
| French (fr) | 0.713 | 0.333 | 0.609 | 0.483 | 0.146 | 0.225 | 0.185 |
| Italian (it) | 0.622 | 0.333 | 0.600 | 0.433 | 0.035 | 0.150 | 0.092 |

Table 3: The multi-lingual performance on TSM (Acc.)

PP-BERT, whose resolved prediction performance $T_r$ is worse than a random guess. This could be attributed to improper perturbation, which misleads the model's predictions. In this case, while its $\Phi_o$ seems to be the best among all other methods, the high $\Phi_r$ and $\Phi$ precisely reflect its poor balance between task performance and decision privacy.

Furthermore, as a common practice in LMaaS, we evaluate the multilingual performance of IoI on four widely used languages: Arabic (ar), Spanish (es), French (fr), and Italian (it). The results in Table 3 demonstrate that IoI is effective in a multilingual setting, with some variation in performance across different languages. On average, IoI achieves the best performance in Italian among all tested languages. However, by adjusting specific settings, IoI can be optimized either for privacy or for task performance.

## 5.3 Analyses

We further study the influence of the technical components described in Section 3.

**Obfuscator Selection.** To verify the loose policy of obfuscator selection that any normal sentence can be a qualified obfuscator, we conduct the following contrastive experiment on SST-2 and SST-5. Specifically, we test the real instances with the obfuscator from the same and different datasets. As shown in Table 4, using instances from a different dataset as obfuscators is indistinguishable from the ones from the same dataset.

**Balancing.** This technique is intended for mitigating the issue of unbalanced obfuscator distributions, thus increasing the accuracy for decision resolution. Here, we study the necessity of balancing. Because SST-5 can be identified as a 5-class classification problem, a unit group $g$ (Equation (8)), in which the obfuscator's labels are uniformly distributed, contains five obfuscators with different labels. According to Equation (9), a group $G_n$ consists of $n$ $g$s. We set the $n$ to be from 1 to 5, that is, 5 to 25 obfuscators. Additionally, as the baseline, we test the obfuscation without balancing by randomly sampling 1 to 25 obfuscators from the obfuscator pool regardless of the classes they belong to. In Figure 6, the performance of applying balancing is presented in a solid line, and the one for the randomly sampled obfuscators is in the dashed line. For the resolved version, without balancing, the accuracy $T_r$ (random) improves by more than 15% from one randomly sampled obfuscator to five, and fluctuates relatively smooth after having more than a unit group of obfuscators (orange dashed line). With balancing $G_n$ ($T_r$, orange solid line), where $n$ ranges from 1 to 5, performs overall better and more steadily than the random samples. Unlike the resolved version, which receives the performance gains, for the obfuscated version, the performance remains stable with different $n$, and is outperformed by the resolved version for more than 10% when the group size is at least a unit (blue lines). As a consequence, balancing archives the best and most robust performance for the resolved version, meanwhile maintaining the maximum gap to the obfuscated version for better decision privacy protection.

**Privacy-Preserving Representation Generation.** PPRG utilizes the compatible input privacy method in a black-box fashion to transform obfuscated instances and obfuscators into representations that preserve privacy. Since the privacy strength and the ability to prevent attacks of the input privacy method are already

| Evaluation | Obfuscator | $T_r$ | $T_o$ |
|:---:|:---:|:---:|:---:|
| SST-2 | SST-2 | 0.907 | 0.770 |
| SST-2 | QNLI | 0.891 | 0.777 |
| SST-5 | SST-5 | 0.467 | 0.339 |
| SST-5 | QNLI | 0.461 | 0.331 |

Table 4: The performance impact with obfuscators from the same and different datasets

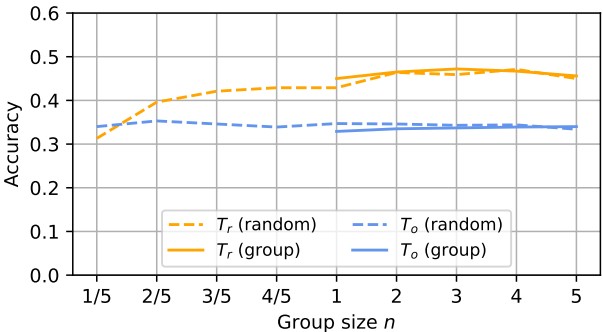

Figure 6: Balancing with different group size (SST-5)

comprehensively studied in its corresponding papers and follow-up works, we focus on the performance of plugging it with IoI.

Table 5 demonstrates the performance of IoI when employing PP-BERT as PPRG. We test PP-BERT with various $\eta$ and compute its accuracy on SST-2. We then use it as PPRG and report $T_r$ and $T_o$. From the observation, the difference between the resolved result $T_r$ and PP-BERT is trivial (see %), which indicates the strong compatibility and recovering ability of IoI. Hence, we conclude that IoI's task performance is dominated by the selected PPRG method as well as input privacy, meanwhile keeping the decision privacy.

**Obfuscated Decision Distribution.** According to Equation (4), the obfuscated decision distribution of $M(b)$ or $M([b;x])$ should be as close to random as possible, and the overall decision distribution of $M(\cdot)$ should also lean towards randomness. For validation purposes, we conduct an additional experiment in Table 6, which presents the decision distribution of $M(b)$, $M([b;x])$ and the overall $M(\cdot)$ on SST-2. Specifically, each cell denotes the distribution of negative/positive decisions. The parameter $k$ controls the strength of the obfuscator, a higher $k$ enhances privacy but reduces utility (details of $k$ is in Appendix B.1).

At $k = 1$ (optimal utility), the distributions of $M(b)$, $M([b;x])$ and $M(\cdot)$ approach randomness. As $k$ increases, the level of randomization intensifies. At $k = 10$, the distribution becomes equivalent to random. Moreover, we set $k = 1$, and report the decision distribution for the other three datasets in Table 7. Similarly, each cell denotes the distribution of decisions. The results show that our method achieves a promising decision distribution even with optimal utility.

**Logit Difference Distribution.** Ideally, after IoI's decision resolution, all true decisions should be correctly recovered. However, some failure cases still occur. According to Equation (12), the dimension with the highest final logit after resolution is selected as the true decision. To better understand the causes of these failures, we analyze the logit difference distribution of decision resolution in SST-2. As shown in Figure 7, for correctly resolved cases, 25.6% of the logit differences exceed 2. In contrast, for incorrect cases, 77.6% of the logit differences are below 2. This observation suggests that many incorrect cases have a small logit difference and could potentially be corrected if the obfuscator were able to nudge the prediction toward the correct dimension slightly more.

**Confidence Distribution.** One potential privacy concern is that the adversary might differentiate between $[b;x]$ and $b$ in input queries based on confidence levels, as obfuscators are preferred to have high confidence in

| PP-BERT      | Acc  | $T_r$ | $T_o$ | %    |
|--------------|------|-------|-------|------|
| $\eta = 200$ | .914 | .912  | .768  | .002 |
| $\eta = 100$ | .925 | .908  | .759  | .018 |
| $\eta = 50$  | .851 | .846  | .698  | .006 |
| $\eta = 25$  | .536 | .528  | .516  | .015 |

Table 5: PPRG-enabled IoI performance (SST-2)

| $k$ | $M([b;x])$ | $M(b)$ | $M(\cdot)$ |
|-----|------------|--------|------------|
| 1   | 0.431/0.569 | 0.5/0.5 | 0.465/0.535 |
| 5   | 0.487/0.513 | 0.5/0.5 | 0.493/0.507 |
| 10  | 0.502/0.498 | 0.5/0.5 | 0.501/0.499 |

Table 6: Decision distribution (SST-2)

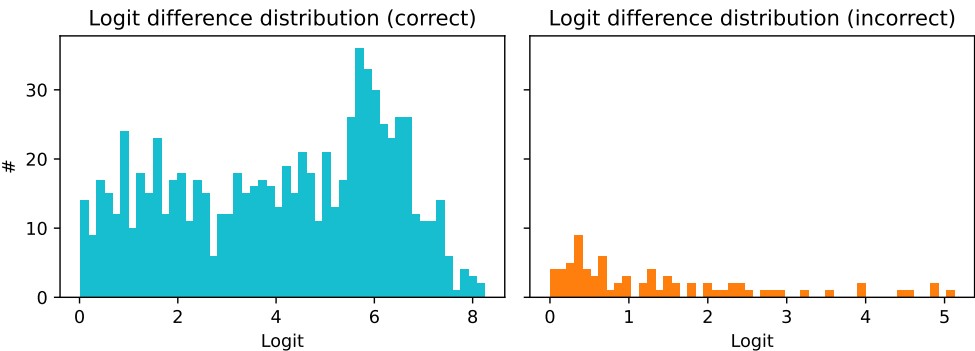

Figure 7: Logit difference distribution of the decision resolution (SST-2)

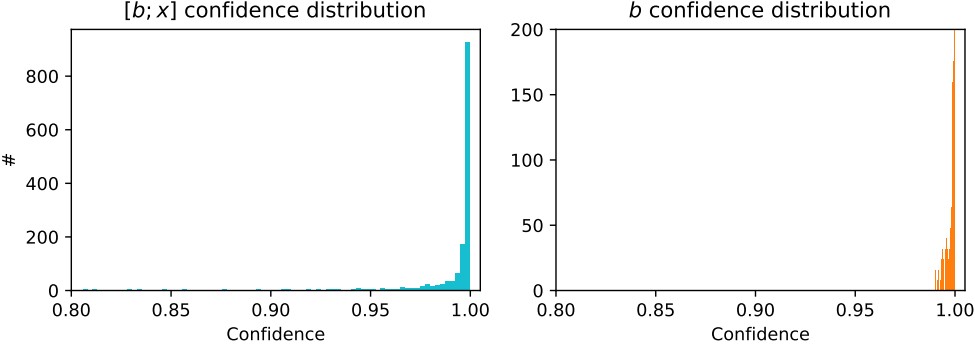

Figure 8: Confidence distribution of $[b;x]$ and $b$ (SST-2)

the decision. However, this does not imply that only high-confidence inferences originate from $b$s. The $[b;x]$ instances can also produce high-confidence results. As shown in Figure 8, for SST-2, we set the minimum confidence threshold for selecting obfuscators to 0.99. Even in this extreme case, the confidence distribution of $[b;x]$ remains predominantly above 0.98, making its distribution nearly indistinguishable from that of $b$s.

**Time Cost.** We conducted an experiment regarding running time (in seconds) on SST-2 and SST-5 with 872 and 1101 instances, respectively. The result is reported in Table 8, where each number averaged from 5 runs. Two numbers in inference are the time cost in the best and worst cases.

| Dataset | $M([b;x])$ | $M(b)$ | $M(\cdot)$ |
|---------|------------|--------|------------|
| SST-5 | 0.141/0.227/0.191 | 0.2/0.2/0.2 | 0.171/0.213/0.195 |
|       | 0.273/0.168 | 0.2/0.2 | 0.236/0.184 |
| MRPC | 0.528/0.472 | 0.5/0.5 | 0.514/0.486 |
| QNLI | 0.532/0.468 | 0.5/0.5 | 0.516/0.484 |

Table 7: Decision distribution ($k = 1$)

| Dataset | Method | Pre-process | Inference | Resolution |
|---------|--------|-------------|-----------|------------|
| SST-2 | Plaintext | 0.002 | 0.936 | - |
| SST-2 | IOI | 0.005 | 1.652/3.327 | 0.060 |
| SST-5 | Plaintext | 0.001 | 1.149 | - |
| SST-5 | IOI | 0.003 | 5.223/10.421 | 0.145 |

Table 8: Time Cost

The results indicate that the experimental outcomes align with the theoretical analysis in Section 4.3 and are even faster in practice, likely due to factors like memory caching. The time cost for decision resolution is also minimal since it involves only summation and averaging operations.

## 6 Related Work

**Privacy Preservation in LMaaS.** Recent studies are actively engaged in addressing the privacy concerns associated with LMaaS. Methods including noise injection (Plant et al., 2021), differential privacy (Hoory et al., 2021; Yue et al., 2021; Xu et al., 2020), and adversarial training (Li et al., 2018; Coavoux et al., 2018), and representation fusion (Zhou et al., 2022) tend to perturb the input text sequence or intermediate representations by reducing unnecessary or sensitive information for PLM's inference. There also exist approaches (Feng et al., 2020; Chen et al., 2022) that seek to protect data flow end-to-end, relying on homomorphic encryption, albeit the execution of such models is very time-consuming and computationally expensive, and needs to modify the model from the server side. On the other hand, to mitigate privacy issues in cloud PLM fine-tuning, offsite-tuning (Xiao et al., 2023) compresses the full PLM into a distilled version that allows users to tune plug-in adapters on their local, which protects the privacy of the user as well as the weights of PLM. Du et al. (2023) exploit local differential privacy to sanitize the embedding (and labels) for fine-tuning. However, none of the above work protects inference decision privacy of the LM under the black-box setting, which is exactly the focus of this work.

**Data Obfuscation.** Although mixup (Zhang et al., 2018) is designed to alleviate the undesirable drawbacks of large deep neural networks, the concept of data combination and its effect on inference with minimal computation overhead is valuable and worth learning. Guo et al. (2019) extend the mixup into the NLP world, and Co-mixup (Kim et al., 2021) discovers the possibility of applying mixup on multiple instances. Besides representation mixup, recent studies also obfuscate authorship of text by neutralizing the stylistic features of text with techniques, such as back-translation or representation disentanglement (Mahmood et al., 2022; Altakrori et al., 2022; Bevendorff et al., 2019). Our instance obfuscated technique is inspired by representation mixup, while representing a pilot approach for LM decision protection.

## 7 Conclusion

In this work, we introduce decision privacy and propose IoI that prohibits information leakage of PLMs under the settings of black-box LMaaS. In contrast to prior works, we consider the end-to-end privacy protection of PLM's input and decision via instance obfuscation. Correspondingly, we define the evaluation metrics tailored for decision privacy and conduct comprehensive experiments regarding task performance and privacy protection. We anticipate our work conveys valuable insight and sheds some light on the trajectory of privacy in NLP.

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

## Limitations

We discuss two main limitations of this work. First, the extra instance inference cost. IoI hides the target instance behind obfuscators so that the target instance never exposes directly to the PLM. To guarantee the strength of privacy protection and stability of the task performance, the strategies, including balancing and randomization, emit additional requests which result in multiple inferences for one instance. However, such incurred cost is not as severe as the previous works, for some of them need significant effort to fine-tune the remote PLMs, and some others require partial/entire model sharing hence compromising the privacy of the model.

Second, IoI is deliberated for text classification tasks in a solely black-box fashion, thus it is not suitable for generative tasks. As for natural language generation, the adaption based on the current method requires addressing the problems such as mix-up tokens and variable lengths of generated text which are non-trivial. We leave this to be the future work.

## Ethical Considerations

Technology innovations generally offer potential benefits, but they also possess the risk of intentional exploitation for nefarious purposes, and LMaaS is not immune to this reality. The presence of regulations and standards establishes a legal structure that ensures responsible utilization of data and grants individuals the right to request the deletion of their data. In the absence of such regulations, society depends on the ethical responsibility of technology practitioners to ensure the ethical usage of data.

Decision privacy, defined in this paper, provides a fundamental direction for protecting the data, as well as LMaaS, from being abusively used. The proposed method technically guarantees the privacy of the input and output data to and from the LMaaS being fully obfuscated. Adopting this method ensures the operations and data are intended for legitimate purposes rather than malicious. Besides, the method itself can seamlessly be integrated into compatible underlying technologies or running systems without any extra modification, which reduces the barriers associated with implementation for increasing accessibility for individuals or organizations.

All experimental datasets used in this work are openly available benchmarks. No demographic or identity characteristics are used in this paper.

---

**Algorithm 1:** Obfuscation generation (GenObfu)

---

**Input:** The number of unit obfuscator groups $n$.
**Output:** The selected obfuscator set $R$.
**Note:** M is the PLM. $C$ is a set of all PLM's labels.

**1 Function GenObfu($n$):**

    // Candidate obfuscator construction
**2**    $L \leftarrow \{c \in C \mapsto []\}$;
**3**    $B \leftarrow$ Random text corpora;
**4**    **for** $b \in B$ **do**
**5**        $c \leftarrow$ M($b$);
**6**        $L[c].append(b)$;

    // Obfuscator selection with balancing
**7**    $R \leftarrow \emptyset$;
**8**    **for** $c \in C$ **do**
**9**        $H \leftarrow$ randomly choose $n$ elements with high inference confidence from $L[c]$;
**10**       $R \leftarrow R \cup H$;
**11**    **return** $R$;

---

# A  Method

## A.1  End-to-End Workflow in Pseudo-code

Algorithm 1 outlines the process for generating obfuscators. In the candidate obfuscator construction phase (Lines 1–6), a mapping is created from the PLM's label to a list, and a set of random text corpora serving as candidates is prepared. These candidates are then categorized based on their inference results. During the candidate selection phase (Lines 7–10), $n$ unit groups of candidates with high PLM decision confidence are randomly chosen.

The end-to-end workflow of IoI in pseudo-code is outlined in Algorithm 2. In Lines 1-9, IoI generates $m \times n$ obfuscators, where each obfuscator group consists of $m$ labels, and there are $n$ such groups. Each obfuscator $b$ is concatenated with $x$ to form $[b; x]$ ($bx$ in the algorithm). In Lines 10-13, both $b$ and the obfuscated instance $[b; x]$ are encoded into privacy-preserving embedding representations using the compatible PPRG method. Before transmitting $BX$ and $B$ to the service provider, the client uniformly shuffles their union into $U$ and records the relationship between each $x$ and its corresponding $[b; x]$s and $b$s in $O$. Lines 10-16 involve sending $U$ to the PLM model for inference. From the service provider's perspective, it cannot differentiate whether an input in $U$ corresponds to $b$ or $[b; x]$. Finally, in Lines 19-23, the raw inference logit set $Z$ is returned to the client. Using $O$, the client reconstructs the associations between $x$ and its corresponding $[b; x]$s and $b$s, then determines the true result $y$ for each $x$ based on Equation (12).

It is unnecessary to run inference repeatedly for all obfuscators. Instead, the results can be precomputed and stored in an obfuscator pool, where each $b$ is associated with its corresponding label. This allows obfuscator results to be reused when needed, reducing redundant computation and communication (highlighted in gray), thereby improving efficiency.

# B  Experiments

We report additional information regarding experiments and analyses in this section.

## B.1  Length Expansion

The privacy strength of a single obfuscated instance $[b; x]$ mainly comes from the domination of $b$. Kim et al. (2021) demonstrate that the model has the ability to map a mixed instance consisting of more than two raw instances to a mixed label. Inspired by that, We expand the length of $b$ to amplify the impact of it on PLM's

---

**Algorithm 2:** IOI workflow

---

**Input:** A set of instances $X$ in the text form. The number of unit obfuscator groups $n$.
**Output:** A set of results $Y$.
**Note:** `GenObfu` is for obfuscator generation. `PPRG` is any compatible PPRG methods. `M` is the PLM. $C$ is a set of all PLM's labels.

    `// Apply obfuscation`
**1** $m \leftarrow |C|$;
**2** $BX \leftarrow \emptyset$;
**3** $B \leftarrow \emptyset$;
**4** **for** $x \in X$ **do**
**5**     $B_i \leftarrow$ `GenObfu`$(n)$;
**6**     $B \leftarrow B \cup B_i$;
**7**     **for** $b \in B_i$ **do**
**8**         $bx \leftarrow b \oplus x$;
**9**         $BX \leftarrow BX \cup \{bx\}$;

    `// Apply PPRG`
**10** **for** $bx \in BX$ **do**
**11**     $BX \leftarrow BX \setminus \{bx\} \cup \{$`PPRG`$(bx)\}$;
**12** **for** $b \in B$ **do**
**13**     $B \leftarrow B \setminus \{b\} \cup \{$`PPRG`$(b)\}$;

    `// Uniform shuffle` $\sigma$
    `// The relation of` $x \in X$ `and its corresponding` $bx$ `and` $b$ `is recorded in` $O$
**14** $U \leftarrow \sigma(BX \cup B)$;

    `// Send` $U$ `to the PLM and run the model`
    `// These steps are executed by the service provider`
**15** $Z \leftarrow \emptyset$;
**16** **for** $u \in U$ **do**
**17**     $z \leftarrow$ `M`$(u)$;
**18**     $Z \leftarrow Z \cup \{z\}$;

    `// Send obfuscated results` $Z$ `back to the client and resolve the true result`
**19** $Y \leftarrow \emptyset$;
**20** **for** $x \in X$ **do**
**21**     $\{(z^{b_j x}, z^{b_j})\}_{j=1}^{m \times n} \leftarrow$ Identify all $bx$ and $b$ pairs associated with $x$ according to $O$;
**22**     $y \leftarrow \arg\max \frac{1}{m \times n} \sum_{j=1}^{m \times n} (z^{b_j x} - z^{b_j})$;
**23**     $Y \leftarrow Y \cup y$;

---

decision by duplicating it $k$ times. [6] Correspondingly, the concatenation sequence length becomes $k|b| + |x|$. Here, we seek to verify the relation between the accuracy and the obfuscator's length over obfuscated and resolved prediction results.

We duplicate $b$ by $k$ times before encoding to realize the length increment, and the dataset we used here is SST-2. As the solid lines shown in Figure 9, $k$ is tested from 1 to 10 consecutively, and it presents a negative correlation to accuracy. Specifically, when $k = 1$, the accuracy of resolved inference is more than 0.9 whereas the accuracy of obfuscated inference is less than 0.8. When $k$ becomes larger, the accuracy of resolved inference $T_r$ drops gradually until it reaches around 0.82 when $k$ is 10. As a comparison, the trend of obfuscated $T_o$ falls quickly almost throughout the $k$'s process and when $k$ is 10, it hits 0.55, which is close to the random guess. The maximum difference of the accuracy between two variants is more than 2.25 times to the minimum difference at the beginning ($k$=2). This experiment demonstrates the massive impact of length expansion on the performance, and a proper $k$ could deviate the obfuscated distribution of PLM's decision far from the true one, thus intensifying the privacy protection. Note that we set another hyper-parameter $n$ in $G_n$ to be 1 and 5 in this experiment; regardless of $n$, the negative correlation holds.

---

[6]The duplication does not hurt the privacy of the generated representation because it is before semantic-neutral shuffling.

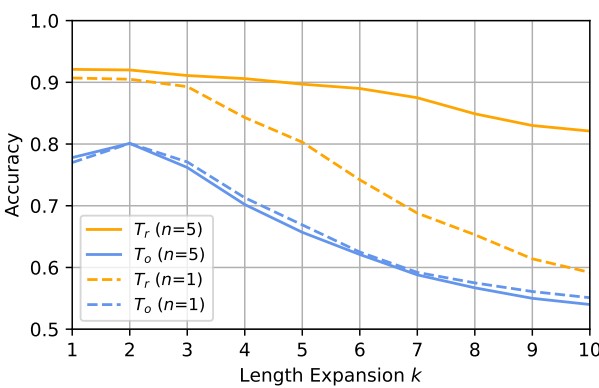

Figure 9: Length Expansion (SST-2)

