# OpenReview forum: "Privacy-Preserving Language Model Inference with Instance Obfuscation"
_TMLR — Rejected by TMLR_

### Review · Reviewer_wBnZ · 2025-02-18

**Summary Of Contributions:**

This paper tackles an important issue of privacy-preserving LMaaS. It proposes a decision privacy method and introduces IoI that prohibits information leakage of PLMs under the settings of black-box LMaaS. In contrast to prior works, this work considers the end-to-end privacy
protection of PLM’s input and decision via instance obfuscation. Experiments are conducted on several datasets.

**Audience:**

Yes

**Claims And Evidence:**

Yes

**Requested Changes:**

Overall, the motivation of this paper is good. I believe a major revision is needed to tackle my following concerns:
1. The case illustration of LMaaS is limited. The authors only provide a single example for illustration. As LMaaS is widely used in multiple downstream tasks in daily life, the authors should provide more detailed discussion and analysis.
2. About the technical designs. It seems that the authors only implement a simple strategy with additional text addition to confuse the attackers. I suggest the authors to provide a comprehensive discussion about their technical contributions. Besides, the design the obfuscator is underexplored. No experiments, ablations, or discussions are provided to analyze this module.
3. Experiments are not convincing. First, the compared methods are out-of-data, which are published before the year 2023. Second, the datasets are insufficient, which can not represent the diverse applications of LMaaS in daily life. Third, no visualization results are provided to illustrate the attack-defense process.
4. Please provide the additional complexity and efficiency costs introduced by the proposed method.
5. Please discuss the failure cases and corresponding limitations.

**Strengths And Weaknesses:**

Strengths
1. The motivation of this paper is good. I think it is an emergency issue in the language model service.
2. This paper is well-written and easy to follow.
3. The case example of the realistic usage is good.

Weaknesses
1. The case illustration of LMaaS is limited. The authors only provide a single example for illustration. As LMaaS is widely used in multiple downstream tasks in daily life, the authors should provide more detailed discussion and analysis.
2. About the technical designs. It seems that the authors only implement a simple strategy with additional text addition to confuse the attackers. I suggest the authors to provide a comprehensive discussion about their technical contributions. Besides, the design the obfuscator is underexplored. No experiments, ablations, or discussions are provided to analyze this module.
3. Experiments are not convincing. First, the compared methods are out-of-data, which are published before the year 2023. Second, the datasets are insufficient, which can not represent the diverse applications of LMaaS in daily life. Third, no visualization results are provided to illustrate the attack-defense process.
4. Please provide the additional complexity and efficiency costs introduced by the proposed method.
5. Please discuss the failure cases and corresponding limitations.

---

> ### Author Response · Authors · 2025-03-13
>
> Thank you for your thorough review and constructive suggestions. We have revised the paper accordingly and highlighted the modifications in blue, covering the introduction (Sec1), methodology (Sec3), privacy proof (Sec4), experiments (Sec5), and the end-to-end pseudo-code (Appendix).
>
> # Responses
>
> - We have included additional practical LMaaS use cases in Sec1, such as personal assistants, customer support chatbots, language translation, and financial advisory.
>
> - (i) We have restated our technical contributions and introduced more mathematical notations to formalize the representation. Specifically, for the contributions of IOI are threefolds:  (1) We explore the feasibility of protecting PLM decisions in a black-box manner, meaning that IOI operates solely during inference without requiring any training, specifically for text classification tasks.  (2) We define decision privacy and comprehensively study instance obfuscation strategies and privacy-preserving decision resolution within this context.  (3) We establish evaluation metrics for decision privacy and empirically validate the performance and privacy strength of the proposed method.
> - (ii) We have completely revised the privacy analysis section (Sec 4). This revision includes defining key terms and security properties, explicitly specifying the adversary as honest-but-curious, and formally distinguishing between the ideal-world functionality and the real-world protocol. We then apply a simulation-based proof to rigorously demonstrate that the proposed method is secure without information leakage.
> - (iii) Regarding the ablation studies on obfuscators (Sec 5.3 and Appendix B.1), we investigated the following:  (1) The selection criteria for obfuscators are highly flexible and practical. (2) Obfuscator balancing enhances the stability of decision resolution.  (3) Obfuscated decisions offer strong privacy protection.  (4) The decision steering effect is controllable through the proportion of obfuscators in the concatenation.
>
> - The baseline methods we selected are all widely used in input privacy research. To the best of our knowledge, no prior studies have specifically addressed LMaaS decision privacy, and consequently, there are no known attacks on decision privacy. Thus, we defined evaluation metrics to measure $T_{o}$, which assesses the obfuscated version of the decision, and $T_{r}$, which evaluates the true (original) performance from the decision resolution.
> - We provided a theoretical analysis of computational and communication overhead in Sec 4.3. Specifically, for $k$ input instances $x$, the total number of requests falls within the range $[(1+k \times n) \times |C|, 2 \times k \times n \times |C|]$, where $n$ is the number of unit groups and $|C|$ represents the number of labels. This analysis demonstrates that IOI is highly efficient with a linear scaling property. In practice, leveraging an obfuscator pool can further reduce redundant computations. Our experiments in Sec 5.3 confirm that the observed time cost aligns with the theoretical analysis, and IOI’s overhead remains minimal.
> - Thank you for this suggestion. We have added an experiment in Sec 5.3 titled “Logit Difference Distribution.” This experiment analyzes the distribution of logit differences between correct and incorrect cases. Specifically, we observed that for correctly resolved cases, 25.6\% of the logit differences exceed 2. In contrast, for incorrect cases, 77.6\% of the logit differences are below 2. This finding suggests that many incorrect cases exhibit small logit differences, indicating that minor adjustments in the obfuscator’s influence could potentially steer the prediction toward the correct class.

---

> > ### Comment · Reviewer_wBnZ · 2025-03-13
> > **Reply to authors' response**
> >
> > I appreciate the authors' feedback, which addresses most of my concerns. However, some minor point still needs to be addressed:
> >
> > 1. The datasets are insufficient, which can not represent the diverse applications of LMaaS in daily life.
> > 2. No visualization results are provided to illustrate the attack-defense process.

---

> > > ### Author Response · Authors · 2025-03-16
> > >
> > > We thank the reviewer's prompt response. We have updated the second revision of the paper and hope our responses address your concerns.
> > >
> > > - We added a comprehensive experiment incorporating baselines and IOI on the financial dataset “Financial PhraseBank.” Additionally, following common practices in LMaaS applications (e.g., chatbots and private advisory), we evaluated IOI’s multilingual performance across four languages (ab, es, fr, and it) using the “Tweet Sentiment Multilingual” dataset. The results from these experiments demonstrate that IOI remains effective across different domains and multilingual settings. These updates are in Sec5.1 and Sec5.2.
> > >
> > > - Complete privacy in the LMaaS life cycle should consider both user input and output. IOI focuses on decision privacy (output) and employs state-of-the-art privacy-preserving representation generation (PPRG) methods to protect input privacy. Since IOI treats LMaaS as a black-box system, the encoding method $E$ serves two roles: (1) transforming the input $x$ into multiple $b$s and $[b;x]$s and (2) encoding $b$s and $[b;x]$s into irreversible representations using PPRG. Thus, $E(x)$ is represented as $\{PPRG([b;x])...\} \cup \{PPRG(b)...\}$ (see refined notation in Section 3.1). The decoding method $D$ resolves all direct (obfuscated) model decisions to their true decisions.
> > >     - **Since PPRG methods have been well studied in their original paper, we do not evaluate them separately or report their attack success rate**. Instead, we assess IOI’s task performance with PPRG enabled, as these methods trade off some task performance for privacy. Specifically, we analyze how PPRG impacts performance when the original $x$ is represented by obfuscated instances. From the ablation studies (Sec5.3), we find that the primary factor affecting task performance degradation is the PPRG method itself, while the performance loss due to IOI is minimal.
> > >     -  **Regarding decision privacy, to the best of our knowledge, no known attack currently exists.** As a contribution of this paper, we propose a set of metrics, $T_{o}$, $T_{r}$, $\Phi_r$, $\Phi_o$, and $\Phi$, to evaluate the effectiveness of privacy protection. These metrics assess how well a method obfuscates LMaaS decisions to appear random while still allowing accurate decoding of the true decision. Specifically, $T_r$ represents the task performance after decision resolution by the data owner, while $T_o$ measures the performance that the adversary (either the service provider or attackers) can retrieve. $\Phi_r$ and $\Phi_o$ quantify how close $T_r$ and $T_o$ are to the baseline and random guessing, respectively, and $\Phi$ balances $\Phi_r$ and $\Phi_o$. Lower values of all three $\Phi$ metrics indicate better task performance and stronger decision privacy protection. Thus, **$T_o$ and $\Phi_o$ can be interpreted as attack-defense performance indicators**.

---

> > > > ### Comment · Reviewer_wBnZ · 2025-03-17
> > > > **Reply to authors' response**
> > > >
> > > > Thanks for your response. I have no more questions. Good Luck!

---

### Review · Reviewer_T5g8 · 2025-03-03

**Summary Of Contributions:**

The paper considers the privacy of querying a pre-trained language model (PLM) 'as a service'. Using text classification as a running example, while one can send a privacy-preserving representation of a query to a PLM, the response of the PLM can still leak sensitive information (a distribution over labels). The authors are interested in coming up with a protocol which allows a user to query a PLM for text classification while protecting both the input and output of the PLM. They define the decision privacy of a protocol as follows: Given an encoding function E and the model M, the distribution of M(E(x)) on input x is close to uniformly random over the n labels. The choice of E has been studied by past work. The authors propose Instance-Obfuscated Inference (IoI) as a protocol. IoI uses a pool of obfuscators, a set of text inputs chosen so that the PLM has high prediction confidence on each input, and each label has an equal number of obfuscators for which the PLM predicts that label given the obfuscator. IoI takes one obfuscator b for each label, and concatenates b to the input x to get an obfuscated example [b; x]. It then sends all [b; x] to the PLM (in shuffled order), as well as all the chosen obfuscators b, as inputs for prediction. The outputs of the PLM are "resolved" by seeing which label's confidence increases the most from having just the obfuscators b as input, to the concatenated examples [b; x] as input, and using this label as the unobfuscated prediction. The authors conduct experiments testing IoI on standard text classification benchmarks such as SST and QNLI and show IoI nearly retrieves the accuracy of the analogous non-private protocol (simply querying the PLM with an unobfuscated input) on most tasks. They do a number of abalations as well, e.g. comparing different methods for choosing the obfuscator sentences, trying an unbalanced distribution of obfuscator labels, and also do a timing comparison of their method and the non-private protocol.

**Audience:**

Yes

**Broader Impact Concerns:**

No concerns.

**Claims And Evidence:**

No

**Requested Changes:**

Critical:
* The problem definition and threat model needs to formalize the restrictions on each step of the protocol, to explain why the above public-key cryptography-based approach is infeasible. If the definition does not rule out this approach, IoI needs to demonstrate some advantage over the above approach. If the definition does rule out this approach (e.g. maybe you are only allowed to interact with the PLM via queries), a compelling argument for the aspect of the problem definition that rules out the approach needs to be included (e.g. why would it be hard to add minimal pre/post-processing by the PLM?).
* Clarify whether the adversary needs to identify the queries from a larger set. If so, this needs to be included in the formalization of the IoI protocol (i.e., how is this set formed, how does the IoI protocol choose to insert its queries into this set). If not, the above criticisms of the privacy proof need to be addressed. In either case, ideally, the threat model/privacy target should be given as a formal definition, and then the claim that IoI satisfies this privacy target should be given as a formal theorem and proof to be more easily verifiable and force the threat model and privacy proof to align.

Minor changes:
* Add pseudo-code for the end-to-end IoI protocol, even if relegated to an appendix.
* Equation (7): This seems imprecise as it's missing a quantifier for $j$. I think what you want is: letting $B_i = \{b: M(b) = c_i \}$, $g$ is a unit group if $\forall i |g \cap B_i| = 1$. And then equation (8) should not be a union but instead say $\forall i |g \cap B_i| = n$.

**Strengths And Weaknesses:**

Strengths:
* Paper studies an important problem, that was ignored by several previous works.
* The empirical results are compelling evidence for the effectiveness of the proposed IoI technique.
* The empirical results are comprehensive and study a number of potential tweaks to the IoI approach and their impact on performance.

Weaknesses:

There are a number of presentation issues, mostly relating to a lack of formalism. Without such formalisms, it is both hard to understand the specific algorithm, threat model, and privacy guarantees of the algorithm. In particular, it seems like there are some issues with the correctness of the privacy proof and degree of improvement over other simple approaches, which are hard to verify due to the presentation.

Some specific examples of presentation issues:
  * Section 3 would greatly benefit from e.g. a pseudo-code block that gives the end-to-end algorithm. Figure 3 only describes a case where one [b; x], b pair is sent to the PLM but it seems fully understanding the method requires understanding how multiple such pairs are chosen and processed.
  * Equation (7) is imprecise; a suggestion for how it should be fixed is given below.
  * "Arbitrary order" needs some precision on page 6; arbitrary usually means any, so giving them in an order that allows the adversary to easily find the pairing but is "arbitrary". It seems like one wants to e.g. uniformly shuffle the order instead.
  * The privacy analysis seems to heavily rely on the fact that the adversary cannot even find the requests a user made in a set of r total requests. However, the threat model doesn't seem to match this - the threat model seems to assume the adversary can find the input/output pairs for the requests made by the user, and that the privacy is handled by the encryption and obfuscation of these input/output pairs. Ideally the threat model should be formalized such that the privacy analysis can be given as some sort of formal theorem and proof based on the threat model to prevent such ambiguities.
  * Related to the above point; the threat model and what types of operations are allowed by the protocol need to be made more precise so that it is clear what other baselines are allowed / not allowed for comparison.

Because the threat model is unclear, the proof of privacy seems problematic. Namely, the threat model as stated in Section 4.1 seems to assume the adversary can see the inputs/outputs sent by a specific user, whereas the privacy proof relies heavily on the adversary's inability to find these requests in a pool of r total requests. Without this, it seems like the privacy breaks down. Specifically, if the threat model allows the adversary to see the input/outputs sent by a specific user, then it seems like the desired claim is that the adversary cannot distinguish the outputs on the encrypted b values from the queries on encrypted [b; x] values (since once the adversary can partition the outputs correctly into b and [b; x] values, they can do decision resolution on their own since the specific pairing of b and [b; x] values is not needed to evaluate (10), since the value of (10) is the same even if one permutes the $c_i^{b_j; x}$ values or $c_i^{b_j}$ values. Hence partitioning the examples into these two sets suffices to violate decision privacy). However, while the encrypted queries are hard to distinguish, the outputs may not be hard to distinguish, especially if the adversary is allowed to know the pool of obfuscators and do their own queries on the obfuscators to learn their outputs. If the adversary is able to do this, they can find all the outputs for the obfuscators, i.e. b values, by just finding the output distributions that match the outputs for their own queries.

In addition, because the restrictions of the threat model and protocol are not made formal, it's unclear why other simple baselines could not succeed. For example, here's a protocol based on public-key cryptopgraphy that doesn't sacrifice any accuracy and gives perfect decision privacy: The PLM generates a public/private key pair. When the user makes a query, they sample a uniformly random permutation of [1, 2, ..., n] and using the public key, encrypts the tuple (permutation, query), and sends this to the PLM. The PLM decrypts the tuple, runs on the query, and then shuffles the distribution of confidence over the labels using the permutation and releases this publicly. The user knows the permutation they chose, so they can invert the shuffle to receive the true label distribution. No accuracy is lost because this recovers the non-private approach. This gives perfect decision privacy the adversary sees a uniformly random shuffle of the confidences, i.e. each label is equally likely to be the argmax after shuffling, and the input encryption is cryptographically secure. Furthermore, the PLM only needs to do a single query (as opposed to the multiple queries IoI requires) and otherwise has minimal overhead compared to the cost of inference. It's not clear what in the paper's problem definition and threat model rules out this approach, which seems to improve on IoI in all aspects.

---

> ### Author Response · Authors · 2025-03-13
>
> Thank you for your detailed review, particularly for your careful examination of the notations and the privacy discussion. We have revised the paper accordingly and highlighted the modifications in blue, covering the introduction (Sec1), methodology (Sec3), privacy proof (Sec4), experiments (Sec5), and the end-to-end pseudo-code (Appendix).
>
> # Responses
> - Pseudo-code: We have added an end-to-end pseudo-code along with detailed explanations in Appendix A.1. Additionally, in Sec3, we added an end-to-end mathematical definition of the encoding function $E$ and the decoding function $D$ (Equation 11 and Equation 13).
> - Equation: Due to the addition of more equations for formal representation, the referenced equation is now Equation 8. The corrected equation is: $g=\{b_j \in B\ |\ M(b_j) = c_i, \forall c_i \in C\}\ \text{with}\ |g|=|C|$. This equation defines a unit obfuscator group $g$ such that for every class $c_i \in C$, there exists exactly one corresponding $b_j \in g$ that maps to $c_i$ via $M(\cdot)$. This guarantees that $g$ is structured to evenly cover all classes, meaning that a $|C|$-size group $g$ contains one qualified combination of $b_j$s. Moreover, we also added and updated other equations in Sec3 to make them clearer and more consistent.
> - Arbitrary order: Thank you for pointing this out. We have revised the wording to consistently use “uniform shuffle” throughout the paper.
> - Security proof: We have completely revised the privacy analysis section (Sec4). This includes defining the relevant terms and security properties, explicitly specifying the adversary as honest-but-curious, and formally distinguishing between the ideal-world functionality and the real-world protocol. We then use a simulation-based proof to rigorously demonstrate that the proposed method is secure without information leakage.
>
> Additionally, regarding the infeasibility of a public-key encryption approach, there are two primary reasons:  (1) IOI is designed for LMaaS in a black-box manner, meaning it does not require any modifications to the deployed LMaaS service, making it applicable in a wide range of scenarios.  (2) General public-key encryption (as well as symmetric encryption) only protects data during transmission but does not prevent the service provider (LMaaS) from storing and learning from the data. Our proposed method is more similar to homomorphic encryption, where only the data owner has knowledge of the real input and output. However, homomorphic encryption or similar techniques require modifications to the LMaaS and introduce significant computational and communication overhead. In contrast, IOI does not require any changes on the LMaaS side and remains highly efficient.

---

> ### Comment · Reviewer_T5g8 · 2025-03-20
> **More questions on privacy**
>
> Thanks for your response and implementing the suggested changes. I have looked at the new pseudocode and privacy proof, and believe the clarity in the paper is greatly increased. I also now understand why the authors are ruling out approaches like public-key cryptography. However, I think my concerns about the decision privacy are not fully addressed. Here's my step-by-step argument for why I think decision privacy may not hold, hopefully the authors can identify which step is wrong.
>
> 1) When the client makes a single query (k=1),  the privacy proof is still meant to hold.
> 2) In this case, the adversary gets to view all $2|G_n|$ decisions y', without any sort of encryption / obfuscation, albeit in shuffled order.
> 3) When k = 1, to do decision resolution it suffices to determine which y' came from obfuscators alone (b) and which came from  padded queries (b;x), in particular one does not need to know the pairing of obfuscators and padded queries. This is because equation (12) is the same for any permutation of the y' corresponding to obfuscators or permutation of y' corresponding to padded queries
> 4) If the set of possible obfuscators (i.e., the support of $B$ in Algorithm 1) is small, the adversary could query the model with all possible obfuscators. Any $y'$ matching the output of the adversary's queries can be assumed to come from obfuscators, so by the previous step the adversary can perform decision resolution on their own. Hence the choice $B$ in Algorithm 1 needs to be specified further to ensure a computationally bounded adversary cannot do this.
> 5) Even ignoring step 4, it is not clear that the task in step 3 above is hard in general. In particular, the obfuscators are chosen such that the model has high confidence responses to them, whereas the outputs on padded queries are likely to be lower confidence (as in the example in Figure 3) Hence, an adversary could guess that the $|G_n|$ values of $y'$ with the highest confidence are the responses to the obfuscators, and have a (perhaps substantially) better than $1 / \binom{2|G_n|}{|G_n|}$ chance of identifying the obfuscators.
> 6) Even if the adversary cannot perfectly identify all the $y'$ that are responses to obfuscator-only queries, identifying even 90% of them might suffice to break decision privacy, as in this case the value inside the argmax in equation (12) can still be approximately correct.

---

> ### Author Response · Authors · 2025-03-21
>
> Thank you again for your suggestion for improving the paper! Here, we clarify the privacy concern you mentioned.
>
> The security of IOI's decision privacy relies on the difficulty of (1) identifying all raw inference results (from $b$s and $[b;x]$s) associated with specific instances $x$s, and (2) distinguishing whether they originate from $b$s or $[b;x]$s.
>
> Let us first discuss Condition (1):
>
> - If there is only a single $x$ (i.e., $k = 1$), the raw input $x$ is represented by $b$s and $[b;x]$s. Equation 12 is used for resolving one $x$, which has $|G_n|$ associated $b$s and $[b;x]$s. The adversary $A$ must determine which results correspond to $b$ and which to $[b;x]$ to perform decision resolution. In this case, Condition (1) becomes trivial if $A$ knows IOI is enabled, as it only needs to solve Condition (2).
>
> - If $k > 1$, $A$ observes $2 \times k \times |G_n|$ instances, meaning it must identify $2 \times |G_n|$ instances out of $2 \times k \times |G_n|$ and determine which $x$ they are associated with (as described in Equation 13). However, since each $b$ and $[b;x]$ is PPRG-encoded, making them irreversible and distinct even for identical inputs, $A$ lacks this knowledge to identify. If $A$ could determine these associations, a stronger PPRG method would be required.
>
> - Unlike the previous bullet points, where we assumed that $b$s are different for every $x$, the number of $b$s does not necessarily have to match the number of $[b;x]$s. A smaller set of $b$s can be used, for example, by employing an obfuscator pool ($P$), which helps reduce computational overhead. However, this does not imply that for $k$ instances of $x$, the client must send $b \in P$ exactly $k \times |G_n|$ times to the service provider. The $b$s and $[b;x]$s do not need to be sent in pairs. As long as the client has $P$ (with all labels precomputed), it only needs to send $[b;x]$s. However, $A$ cannot determine whether a given request consists solely of $[b;x]$s. Furthermore, $A$ has no access to $P$, so it remains unaware of the total number of available $b$s.
>
> - Additionally, although not discussed in the paper for simplicity, the client could send distracting instances (which are normal inputs unassociated with any $x$), or intermix $b$s and $[b;x]$s within other non-private queries. This strategy further increases the difficulty for $A$ in distinguishing between $b$s and $[b;x]$s.
>
> For Condition (2):
>
> - While high-confidence $b$s are preferred, it is not the case that only high-confidence inferences originate from $b$s. The $[b;x]$ instances can also yield high-confidence results. We added an experiments named Confidence Distribution (Sec5.3) to compare the confidence distribution of $[b;x]$ and $b$'s decision on SST-2. We set the minimum confidence threshold for selecting obfuscator $b$ to 0.99. Even in this extreme case, the confidence distribution of $[b;x]$ remains predominantly above 0.98, making its distribution nearly indistinguishable from that of $b$s.
>
> - Choosing high-/low-confidence $b$s presents a trade-off between performance and privacy. High-confidence $b$s generally better steer the decision distribution but may also be easier to identify as $b$s.
>
> - If distracting high-confidence instances are also introduced, identifying $b$s becomes even more challenging.
>
>
> Therefore, satisfying both Condition (1) and Condition (2) simultaneously is challenging for a computationally bounded adversary $A$. Moreover, even if $A$ successfully identifies some, but not all, of the $b$s and $[b;x]$s associated with a given $x$, there is no guarantee of correct decision inference. This is why we employ a group of obfuscators to ensure performance.
>
>
> Additionally, we clarify that IOI-encoded instances are indistinguishable from normal (non-private) instances, as they are all embeddings of natural text, unless the selected PPRG method introduces some detectable "signature." Thus, $A$ should not aware that IOI is enabled or not.

---

> ### Comment · Reviewer_T5g8 · 2025-03-24
>
> Thanks for your response. I appreciate that the authors have also added experiments to try to validate whether some of these attacks would succeed or not. However, I was under the impression the authors had a formal proof of privacy, but based on the discussion the privacy guarantee seems to be based on heuristic arguments instead. For example, statements like these in the response are informal and concerning:
>
> * Even in this extreme case, the confidence distribution of $[b;x]$ remains predominantly above 0.98, making its distribution nearly indistinguishable from that of $b$s."
>   * This is a single example, and even then 'nearly indistinguishable' is not formalized here. It's not clear why this indistinguishability suffices to prevent an adversary from correctly using decision resolution, and it's also possible in other data domains the distributions are more distinguishable.
> * "Moreover, even if $A$ successfully identifies some, but not all, of the $b$s and $[b;x]$s associated with a given $x$, there is no guarantee of correct decision inference."
>   * There is no guarantee, but that is not enough. First, it is your responsibility to prove that this strictly fails in some sense, not just that "there is no guarantee". Second, even if this attack fails, there could be other attacks we haven't thought of that you also need to rule out.
> * "the client could send distracting instances"
>   * It seems either (i) this is critical for privacy/security and needs to be included in the protocol and proof, or (ii) the authors believe it's not critical and this is a moot point, since you should provide some formal privacy in all cases.
>
> My goal with these responses is not that I want the authors to respond to these individually (after all, for every attack you can qualitatively argue fails, someone can come up with a new attack), but I want to demonstrate why the heuristic/qualitative answers to the proposed attacks aren't compelling. Instead, I was hoping the authors could show that they ruled out these weaknesses preemptively, by pointing to some part of your proof. e.g. with a formal proof you should have been able to respond to bullet 6 in my previous response with something like "we proved that exactly distinguishing the padded examples from the obfuscators is strictly necessary for decision resolution and thus identifying a constant fraction of the padded examples correctly does not allow non-trivial improvements in decision resolution", rather than "there is no guarantee (but, it might work in some cases)"
>
> There are many high-profile instances of proposals for privacy/security-enhancing techniques which qualitatively seemed secure and maybe even had initial strong heuristic performance but no formal guarantees, and then attacks later being found to break the privacy/security (e.g.; the de-anonymization of the Netflix dataset in https://www.cs.utexas.edu/~shmat/shmat_oak08netflix.pdf, Carlini et al.'s attack on InstaHide in https://arxiv.org/abs/2011.05315). In some cases, data was released after being processed by the technique, and hence compromised (e.g. Honig et al.'s attack on Gleam in https://arxiv.org/abs/2406.12027). For this reason, I think that it is the authors' responsibility to show a formal privacy guarantee that rigorously preemptively rules out every possible attack (rather than just qualitatively explaining why the attacks), and it seems based on the discussion that the current paper fails to do so.

---

> ### Author Response · Authors · 2025-03-26
>
> Thank you for your response.
>
> We would like to clarify that the informal explanations and experimental results were provided for ease of understanding. Our approach does not rely on simple hashing or removing personally identifiable information (PII). The security of our protocol is established through simulation-based proof.
>
> To address the inherent uncertainty of language models (arising from their probabilistic nature, data ambiguity, overparameterization, prompt sensitivity, decoding strategies, etc), we introduce a parameterized security definition (Sec4, Definition 5) that quantifies this uncertainty in relation to security by bounding the difference in the inference logit distribution using KL divergence. Our proof shows that, under a given security parameter, the adversary gains no information beyond random strings when executing the protocol.
>
> Regarding the "distracting instances," we did not include them in the formal proof because, within a simulation-based security framework, their presence does not alter the adversary's knowledge. The adversary still learns nothing from the protocol execution. The mention of distracting instances was to illustrate that mixing inputs with additional distracting instances makes it even more difficult for an adversary to exhaust all possibilities. Specifically, if the adversary originally needed to identify $m$ instances from a set of $n$, adding $k$ distracting instances increases the search space from ${m}\choose{n}$ to ${m}\choose{n+k}$. However, this does not affect the security proof, as the adversary still perceives only random values. Analogous to a cryptographic system, this is equivalent to increasing the password size.
>
> Furthermore, we note that it is not feasible to preemptively rule out every possible attack, even for the most advanced cryptographic systems. Cryptographic security relies on the computational hardness of certain mathematical problems, making it infeasible for a computationally bounded adversary to break the system. However, if a new attack is developed that exploits vulnerabilities in a cryptographic scheme, it could reveal additional information beyond brute-force enumeration, and such attacks cannot be predicted or formally ruled out in advance.

---

### Review · Reviewer_U6bN · 2025-03-04

**Summary Of Contributions:**

The paper deals with privacy-preserving methods in the following scenario: a user sends an obfuscated input to a cloud to perform inference on a specific task, the cloud sends back an obfuscated output, the user should be able to infer the model's real output from the obfuscated output. However, nor the cloud provider, nor a potential eavesdropper should be able able to recover information about either the real input nor the real output.

Whereas methods for obfuscating the input (privacy-preserving text representation) are abundant, the focus on the privacy of the decision is somewhat new and is the main contribution of the paper.

After describing and conceptualizing this scenario in mathematical terms (section 2) , the paper introduces a method called IoI for Instance-Obfuscated Inference (section 3). The method consists in concatenating the true input x to randomly chosen examples {b_1, ...} for which the model's output (distribution over output classes) is known to the user.
Then the concatenated examples are passed through a privacy-preserving representation method and sent to the cloud-based model.
The model returns an output distribution y'. The user can infer the true distribution y for x, based on the hypothesis that y' aggregates the output distribution of the whole input set (x + obfuscators) and from the knowledge of the output distributions for individual obfuscators (which the attacker does not have).

The proposed method is evaluated (section 5) on utility (accuracy on task) and decision privacy, on a set of text classification tasks (sentiment analysis, paraphrase detection, NLI) and is shown to protect decision privacy more efficiently than baselines, while retaining high end-task accuracy.

**Audience:**

Yes

**Broader Impact Concerns:**

I do not have ethical concerns regarding the research described in this paper.

**Claims And Evidence:**

Yes

**Requested Changes:**

- the paper only experiments with English, but does not disclose it. Since this is a strong limitation of the scope of the paper, it should be acknowledged early (introduction of the paper) that "language" / "text" refer in fact to written English.

- rephrase or clarify the argument about mixup (see weaknesses box)

- it was unclear (until section 5.2) whether the proposed method is only used at inference time or requires training, I would suggest to state it much earlier in the paper.

- is the prediction for a single example stable across multiple obfuscations?

- please specify how the hyperparameters presented in Table 2 were chosen

**Strengths And Weaknesses:**

strengths:

- the conceptualization of decision privacy is clear and helpful, this is a contribution in itself.
- as far as I can tell the proposed method is sound and achieves good results as regards decision privacy compared to baselines. The proposed evaluation measures are meaningful and easy to interpret.
- the paper features an extensive set of analysis experiments to assess the behaviour of the model (in particular with varying number of obfuscators) and reports on the full tradeoff privacy/accuracy/computational cost

weaknesses:

- the method is based on a strong hypothesis: namely that the output distribution of the model for the concatenation of two examples is the average of the output distributions on individual examples. this is justified as: "The instance obfuscation is motivated by mixup (Zhang et al., 2018), originally proposed for data augmentation. Zhang et al. (2018) theoretically shows that mixup can produce virtual feature-target pairs sampled from the same vicinal distribution as the original data." This point needs clarifications: Zhang et al uses fixed-size data points and combines them as x = (l * x1 + (1-l) * x2), rather than with a concatenation. Moreover their argument seems really empirical rather than theoretical (unless I missed something). The fact that this is a good data augmentation method does not imply that a model trained without mixup will have the same properties. I think this motivation should be rephrased accordingly.

- The tasks are really simple (classification tasks with very small label sets), which limits the general applicability of the method since (i) LLM as a service is increasingly used for generation tasks (ii) tasks with large label sets require more obfuscators and lead to much slower inference (inference time scales linearly with the number of obfuscators). The computational overhead to safeguard privacy might be quickly too high for IoI.

- The paper does not mention a code release.

- I think it would have been important to evaluate also the privacy of the input representations since IoI as I understood it might also improve input privacy.

---

> ### Author Response · Authors · 2025-03-13
>
> Thank you for your thorough review and constructive suggestions. We have revised the paper accordingly and highlighted the modifications in blue, covering the introduction (Sec1), methodology (Sec3), privacy proof (Sec4), experiments (Sec5), and the end-to-end pseudo-code (Appendix).
>
> # Response to weaknesses
> - Thank you for pointing this out. Mixup is primarily an empirical study with some theoretical justifications. We updated the content in Sec3.1 to explain how our method is motivated by the mixup and why we chose concatenation. Specifically, our initial experiments indicate that constructing $E(\cdot)$ by directly mixing embedding vectors of $x$ and $b$, following the original approach of mixup, results in unstable performance for $D(\cdot)$. This instability arises because a vanilla PLM without mixup fine-tuning may lack the ability to accurately infer the mixed label solely from the mixed input. Therefore, we replace the proportional mixup of $b$ and $x$ with concatenation. Additionally, we have introduced mathematical notations to formally define how a query $x$ is represented by $[b;x]$ and $b$.
> - As a pilot study, we haven’t explored the feasibility of using the proposed method in generative tasks. Directly extending the current approach token by token in the generative tasks without optimization may introduce huge computational overhead. However, for the English text classification tasks our method is designed for, this overhead has been optimized. As analyzed in Sec4.3, the communication cost can be reduced to $(1+k \times n) \times |C|$ instead of $2 \times k \times n \times |C|$ when leveraging an obfuscator pool for efficiency. Our experiments in Sec5.3 further confirm that the observed time cost aligns with the theoretical analysis.
> - We will release the code once the paper gets accepted.
> - LM-based embeddings without any PPRG method applied remain vulnerable to inverse attacks. The IOI method does not alter embeddings but instead transmits the embeddings of $b$ and $[b;x]$ rather than sending $x$ directly. Consequently, while the embedding representations of $b$ and $[b;x]$ differ from $x$, they are still simple LM-based embeddings and remain susceptible to inversion attacks. Therefore, we employ PPRG to protect the privacy of input embeddings. We have revised the privacy analysis in Sec4 and now provide a rigorous security proof using a simulation-based approach.
>
> # Response to request changes
> - We added an experiment for evaluating IOI’s multilingual performance across four languages (ab, es, fr, and it) using the “Tweet Sentiment Multilingual” dataset. The results from these experiments demonstrate that IOI remains effective with multilingual settings. Please see Sec5.1 and Sec5.2.
> - See “Response to weaknesses”.
> - We added a clarification that our proposed method operates solely during inference without requiring any training in Sec1.
> - Yes, for a given $x$, any qualified group of obfuscators can be used for obfuscation, regardless of the specific obfuscator instance chosen. As described in the balancing method (defined in Sec3.1 and analyzed in the ablation study in Sec5.3), a larger obfuscator group generally leads to better and more stable performance in decision resolution.
> - Are you referring to the hyper-parameters in Table 1? Table 2 lists the parameters used for all experiments reported in Table 1.

---

> > ### Comment · Reviewer_U6bN · 2025-03-20
> >
> > thanks and sorry for the late reply!
> > - about mulitlinguality: my main point was to disclose the language you are working on, but multiligual experiments are even better!
> > - thank you for all the clarifications, those cleared most of my misunderstandings!
> > - hyperparameters: I wondered how the 'min confidence for b' was chosen, high values make sense but why 0.9 to 0.99 depending on the dataset? were they chosen from experiments on a validation set? or calibrated on the test set?

---

> > > ### Author Response · Authors · 2025-03-20
> > >
> > > Thank you for your response! To answer your hyper-parameter question:
> > >
> > > The confidence of $b$ depends on the dataset. Generally, a high-confidence $b$ is preferred because it has a stronger influence on the decision distribution (as discussed in Sec3.1). However, in some cases, very few or no candidates satisfy an extremely high confidence threshold (e.g., 0.99). To ensure a sufficient number of available $b$s, we lower the minimum confidence threshold accordingly. In practice, $b$s can be selected either from the same dataset (e.g., by pre-splitting the training set) or from a different dataset. As the empirical study in Sec5.3, IOI has a flexible obfuscator selection policy and is not sensitive to obfuscators from different datasets.

---

### Decision · Action_Editor_9teQ · 2025-04-04

**Recommendation:** Reject

**Comment:**

-

**Audience:**

People in the area of privacy.

**Claims And Evidence:**

This paper has had quite a vigorous discussion with one of the reviewers, and more discussion with the others. After carefully considering it with one of the reviewers and further discussing it with them, I believe this paper is not yet ready for publication. The main criterion for TMLR is soundness and accuracy (https://jmlr.org/tmlr/acceptance-criteria.html), and I think the burden of proof for that is still on the author's shoulders. Please note that the reviewer who with the concerns is an expert in this area. Their main concern is that the definitions in section 2.1 do not match the rest of the theory and the proofs of decision privacy.

More specifically, the reviewer states in our discussion: "[regarding privacy...] they define it in Section 2.1, state their goal is to achieve it in Section 2.2, and then in Section 4.2 claim to prove their protocol satisfies decision privacy, but based on the discussion with the authors I do not believe it does satisfy decision privacy."

In addition, the reviewer states: "I do not believe there is a clear route to achieving a formal proof. I proposed several ways one might start to break decision privacy, and the authors' responses were qualitative/heuristic suggestions for why these wouldn't work. [...] didn't find these particularly convincing, [...] some of the suggested techniques can still lead to a break in privacy and hence decision privacy does not hold for their technique and hence there is no route to a proof."

**Resubmission Of Major Revision:**

The authors may consider submitting a major revision at a later time.